# Gut microbiota impact on the peripheral immune response in non-alcoholic fatty liver disease related hepatocellular carcinoma

Jason Behary[1,2,3], Nadia Amorim[1,2], Xiao-Tao Jiang[1,2], Anita Raposo[1,2], Lan Gong [1,2], Emily McGovern[1,2], Ragy Ibrahim[3], Francis Chu[4], Carlie Stephens[3], Hazem Jebeili[3], Vincenzo Fragomeli[5], Yen Chin Koay[6], Miriam Jackson[1,2], John O'Sullivan[6], Martin Weltman[5], Geoffrey McCaughan[7,8], Emad El-Omar [1,2,3] & Amany Zekry[1,2,3✉]

The gut microbiota is reported to modulate the immune response in hepatocellular carcinoma (HCC). Here, we employ metagenomic and metabolomic studies to characterise gut microbiota in patients with non-alcoholic fatty liver disease (NAFLD) related cirrhosis, with or without HCC, and evaluate its effect on the peripheral immune response in an ex vivo model. We find that dysbiosis characterises the microbiota of patients with NAFLD-cirrhosis, with compositional and functional shifts occurring with HCC development. Gene function of the microbiota in NAFLD-HCC supports short chain fatty acid production, and this is confirmed by metabolomic studies. Ex vivo studies show that bacterial extracts from the NAFLD-HCC microbiota, but not from the control groups, elicit a T cell immunosuppressive phenotype, characterised by expansion of regulatory T cells and attenuation of CD8 + T cells. Our study suggest that the gut microbiota in NAFLD-HCC is characterised by a distinctive microbiome/ metabolomic profile, and can modulate the peripheral immune response.

[1] St George and Sutherland Clinical School, University of New South Wales, Sydney, Australia. [2] Microbiome Research Centre, St George and Sutherland Clinical School, University of New South Wales, Sydney, Australia. [3] Department of Gastroenterology and Hepatology, St George Hospital, Sydney, Australia. [4] Department of Surgery, St George Hospital, Sydney, Australia. [5] Department of Gastroenterology and Hepatology, Nepean Hospital, Sydney, Australia. [6] Charles Perkins Centre, Faculty of Medicine and Health, University of Sydney, Sydney, Australia. [7] Liver Injury and Cancer, Centenary Institute, Royal Prince Alfred Hospital, Sydney, Australia. [8] AW Morrow Gastroenterology and Liver Centre, Royal Prince Alfred Hospital, Sydney, Australia. ✉email: a.zekry@unsw. edu.au

Hepatocellular carcinoma (HCC) is the fourth leading cause of cancer-related mortality worldwide[1]. Chronic liver inflammation and liver cirrhosis are key to HCC development, with common risk factors including viral hepatitis, alcoholic liver disease, and non-alcoholic fatty liver disease (NAFLD). The global pandemic of obesity and type II diabetes mellitus has seen liver cirrhosis due to NAFLD (NAFLD-cirrhosis) become the most common liver disease worldwide, with NAFLD-related HCC (NAFLD-HCC) projected to be the main cause of HCC[2,3].

Emerging data support a microbiome signature that defines liver cirrhosis and correlates with markers of liver disease severity[4–7]. Human studies are now attempting to define HCC microbial and metabolite signatures[8,9] whilst animal studies have implicated the gut microbiota and its metabolites in HCC pathogenesis through various mechanisms related to peripheral and intrahepatic inflammatory and immune responses[10–12].

Increasing evidence suggests that the peripheral immune response in HCC influences the clinical course of the disease, response to therapies and overall survival. To this effect, an increased number of tumour-associated antigen-specific T cells in the peripheral blood of HCC patients is associated with a significantly lower risk of HCC recurrence following radiofrequency ablation[13]. Also, relevantly, in the field of cancer immunotherapeutic research, elucidating immunosuppressive mechanisms promoted by cancer cells has become vital to developing therapeutic strategies. In this regard, regulatory T cells (Tregs), are among those cells that suppress host antitumor immunity. Tregs are increased in the peripheral blood of HCC patients, and are associated with tumour progression[14–17], mainly by impairing cytotoxic CD8+ T cell function[17].

In parallel with these important data, there is mounting evidence that the gut microbiota can modulate T-cell immunity both directly and through metabolites, including short-chain fatty acids (SCFAs)[18–20]. SCFAs, in particular butyrate, have important immunomodulatory functions[20–22].

Taken together, understanding factors that influence the immune response in HCC offers a unique opportunity to develop both prognostic markers and new therapeutic approaches. Therefore, we aim to define the composition of the gut microbiota in NAFLD-related liver cirrhosis with and without HCC, and examine its effect on the peripheral immune response. We first employ shotgun metagenomics and metabolomics to define the structure, function and metabolomic capability of the gut microbiota in patients with NAFLD-cirrhosis, with and without HCC compared to non-NAFLD controls. Next, we utilise a well described ex vivo cell culture model[23] to examine the effect of NAFLD-HCC-associated microbiota and metabolome on the peripheral immune response.

Here we report that dysbiosis characterises the microbiome of patients with cirrhosis (with or without HCC), with distinct compositional and functional shifts occurring with the development of HCC. Gene function of the microbiome in NAFLD-HCC support short chain SCFA production. Ex vivo studies confirm that the composition of the gut microbiota in NAFLD-HCC promote an immunosuppressive milieu, which is HCC, rather than liver cirrhosis specific. The data support the notion that the gut microbiota in NAFLD-HCC can be exploited to develop prognostic markers, and plan gut modulating therapeutic strategies.

## Results

**Baseline characteristics of the study cohort.** A total of 90 subjects were recruited in the study; 32 with NAFLD-HCC, 28 with NAFLD-cirrhosis and 30 non-NAFLD control. All groups were matched for age, gender and BMI (Table 1).

All subjects NAFLD-HCC and NAFLD-cirrhosis had transient elastography scores consistent with liver cirrhosis (Table 1). Subjects with NAFLD-HCC and NAFLD-cirrhosis were matched for severity of liver disease based on Child-Pugh and Model for End-Stage Liver Disease-Sodium (MELD-Na) scores (Table 1). There was no difference in liver biochemistry in NAFLD-HCC and NAFLD-cirrhosis groups, and no evidence of portal hypertension clinically, radiologically, or at endoscopy in the NAFLD-cirrhosis group or at pre-operative HVPG in the NAFLD-HCC group.

**Table 1 Baseline clinical and pathological variables of study cohort.**

| Total (n = 90) | Non-NAFLD control (n = 30) | NAFLD-cirrhosis (n = 28) | NAFLD-HCC (n = 32) | P-value (3 group comparison) |
|---|---|---|---|---|
| Age (years) | 61.2 (±7.2) | 62.6 (±10.2) | 65.0 (±6.2) | ns |
| Gender (M) | 80% | 78.6% | 78.1% | ns |
| BMI (kg/m$^2$) | 29.7 (±4.3) | 33.6 (±6.4) | 31.5 (±7.6) | ns |
| Transient elastography score (median kilopascals (kPa)) | – | 14.2 (±4.3) | 15.1 (±4.6) | ns |
| Child Pugh score | – | 5.1 (±0.3) | 5.1 (±0.4) | ns |
| MELD-Na score | – | 12.1 (±0.6) | 12.1 (±0.5) | ns |
| Varices (%) (radiology or endoscopy) | – | 0 | 0 | ns |
| Type II diabetes mellitus (%) | 13.3 | 50* | 50* | 0.0033 |
| Metformin (%) | 10 | 39.3* | 37.5* | 0.0199 |
| Essential hypertension (%) | 16.6 | 57.1* | 46.9* | 0.0044 |
| Dyslipidaemia (%) | 10 | 39.3* | 37.5* | 0.0199 |
| Diet | Ad libitum diet | Ad libitum diet | Ad libitum diet | |
| ALT (U/L) | 25.4 (±20.9) | 41.4 (±16.8)* | 45.8 (±24.0)* | 0.0006 |
| AST (U/L) | 29.2 (±17.4) | 48.6 (±22.9)* | 54.8 (±39.2)* | 0.0019 |
| Platelet (x10$^9$/L) | 283.7 (±51.2) | 193.5 (±48.5)* | 192.2 (±111.2)* | <0.0001 |
| Albumin (g/L) | 40.2 (±5.0) | 40.0 (±2.8) | 38.8 (±5.4) | ns |
| International normalised ratio (INR) | 0.98 (±0.1) | 1.03 (±0.1) | 1.07 (±0.2) | ns |

Values are represented as mean (±SD) or %. P values calculated by one-way ANOVA with Tukey's post hoc test for continuous variables and Chi-squared and Fisher's exact test for binary variables. P < 0.05 considered statistically significant. No significant difference was seen across all variables between the NAFLD-cirrhosis and NAFLD-HCC groups.
BMI body mass index, MELD-Na model for end-stage liver disease-sodium, ALT alanine aminotransferase, AST aspartate aminotransferase, INR international normalised ratio, ns not statistically significant.
*Indicates P < 0.05 compared to non-NAFLD control group (2 group comparison).

**Table 2 Tumour characteristics in NAFLD-HCC group.**

| Total (n = 32) | NAFLD-HCC (n = 32) |
|---|---|
| Tumour number | 1.06 (±0.2) |
| Tumour size (cm) | 2.68 (±1.45) |
| BCLC stage (%) | Stage 0 (84%), Stage A (16%) |
| Histology confirming cirrhosis (%) | 100% |
| Tumour grade (well differentiated, %) | 78.1% |
| HVPG (mmHg) | 5 (±1.8) |
| Alpha-fetoprotein (ng/mL) | 24 (±6) |

Values are represented as mean (±SD) or %.
BCLC stage Barcelona Clinic Liver Cancer Stage, HVPG hepatic venous pressure gradient.

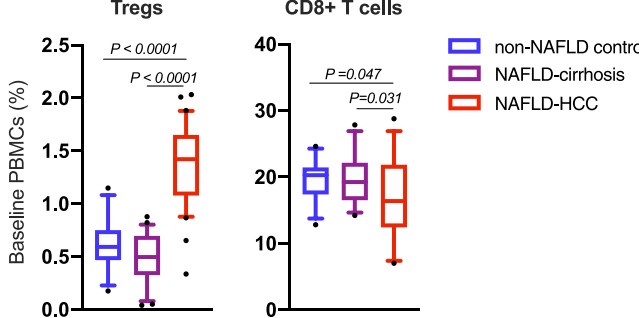

**Fig. 1 Subjects with NAFLD-HCC have a peripheral immunosuppressive profile at baseline compared to NAFLD-cirrhosis and non-NAFLD controls.** Quantification of regulatory T cells (Tregs; CD3+ CD4+ CD25+ Foxp3+) and CD8+ T cells (CD3+ CD8+) in peripheral blood mononuclear cells (PBMCs) from non-NAFLD control, NAFLD-cirrhosis, and NAFLD-HCC subjects at baseline. Sample size is n = 30 non-NAFLD control, n = 28 NAFLD-cirrhosis, n = 32 NAFLD-HCC as biologically independent samples. Data represented as % of CD3+ lymphocytes. Box plots indicate median (middle line), 25th, 75th percentile (box) and 5th and 95th percentile (whiskers) as well as outliers (single points). P values are calculated by one-way ANOVA for 3 group comparison and Tukey's test for 2 group comparison. Detailed data is shown in Supplementary Table 1. Source data are provided as a Source Data file.

There was no difference in the prevalence of type II diabetes, essential hypertension, dyslipidaemia or metformin use in NAFLD-HCC and NAFLD-cirrhosis groups (Table 1). However, subjects with liver disease (NAFLD-HCC or NAFLD-cirrhosis) had a higher prevalence of metabolic disease (type II diabetes, essential hypertension and dyslipidaemia), metformin use and liver biochemistry than non-NAFLD controls (Table 1).

All subjects with NAFLD-HCC underwent surgical resection (with curative intent) for the treatment of HCC and had confirmed cirrhosis and HCC on histology. Additional relevant tumour characteristics including tumour number, size, Barcelona Clinic Liver Cancer (BCLC) stage, tumour grade, pre-operative HVPG, and alpha-fetoprotein level are shown in Table 2.

**Baseline immune cell quantification in peripheral blood.** At baseline, NAFLD-HCC subjects had a higher percentage of regulatory T cells (Tregs; CD3+CD4+CD25+Foxp3+) compared to NAFLD-cirrhosis and non-NAFLD controls (P < 0.0001) (Fig. 1). Conversely, NAFLD-HCC subjects had a lower percentage of CD8+ T cells (CD3+ CD8+) compared to NAFLD-cirrhosis and non-NAFLD controls (P = 0.017) (Fig. 1). No difference in percentage of Tregs or CD8+ T cells was seen between NAFLD-cirrhosis and non-NAFLD control at baseline (Fig. 1).

Detailed baseline immune data including overall, and two-group comparisons are presented in Supplementary Table 1.

**Shotgun metagenomic sequencing confirmed dysbiosis in liver disease and identified a consortium of SCFA-producing bacteria in NAFLD-HCC.** Shotgun metagenomic sequencing of faecal samples confirmed dysbiosis in NAFLD-HCC and NAFLD-cirrhosis compared to non-NAFLD controls (Fig. 2a, b). Subjects with NAFLD-HCC and NAFLD-cirrhosis had reduced α-diversity indices; observed number of species (Fig. 2a) and Chao-1-richness index (Supplementary Fig. 1), compared to non-NAFLD controls. No difference in other α-diversity measures, such as Shannon's diversity index or Evenness index was seen between groups (Supplementary Fig. 1). β-diversity as determined by constraint analysis of principle (CAP) when controlling for age and gender was significantly different between NAFLD-HCC and NAFLD-cirrhosis compared to non-NAFLD control groups (P = 0.004) (Fig. 2b).

At the phylum level, NAFLD-HCC was characterised by expansion of Proteobacteria compared to non-NAFLD controls (P = 0.041), but not compared to NAFLD-cirrhosis (Fig. 2c). No other differences were seen between groups at the phylum level.

However, at the family level, expansion of Enterobacteriaceae was observed in NAFLD-HCC compared to NAFLD-cirrhosis (P = 0.033) and non-NAFLD controls (P = 0.025). NAFLD-HCC was also characterised by a reduction in Oscillospiraceae (P = 0.038) and Erysipelotrichaceae (P = 0.025), compared to non-NAFLD controls but not compared to NAFLD-cirrhosis (Fig. 2d).

Additionally, changes in microbiome composition at the family level were seen in the NAFLD-cirrhosis cohort. NAFLD-cirrhosis was characterised by an expansion of Eubacteriaceae, compared to both NAFLD-HCC (P = 0.013) and non-NAFLD controls (P = 0.0002) (Fig. 2d). Furthermore, expansion of Coriobacteriaceae (P = 0.034) and a reduction in several Bacteroidetes: Muribaculaceae (P = 0.0003), Odoribacteraceae (P = 0.018), and Prevotellaceae (P = 0.009) were observed in NAFLD-cirrhosis compared to non-NAFLD controls but not compared to NAFLD-HCC (Fig. 2d).

Examination of the microbiome at the species level identified a consortium of common bacterial species significantly enriched in both NAFLD-HCC and NAFLD-cirrhosis compared to non-NAFLD controls (Fig. 2e and Supplementary Fig. 2). These included *Bacteroides xylanisolvens* (P = 0.008), *Ruminococcus gnavus* (P < 0.0001), and *Clostridium bolteae* (P < 0.0001) (Fig. 2d). Importantly, two species were found to discriminate NAFLD-HCC from NAFLD cirrhosis; *Bacteroides caecimuris* (P < 0.0001) and *Veillonella parvula* (P = 0.002), were both significantly enriched in NAFLD-HCC, compared to NAFLD-cirrhosis and non-NAFLD controls (Fig. 2e). This finding was also seen at linear discriminant analysis of effect size (LEfSe), thereby confirming that both *B. caecimuris* and *V. parvula* are specifically enriched in NAFLD-HCC compared to NAFLD-cirrhosis and non-NAFLD controls (LDA log$_{10}$ 2.76 and 3.37, respectively) (Supplementary Fig. 3). Therefore collectively, NAFLD-HCC microbiome was enriched in five bacterial species (Fig. 2e) previously reported to be SCFA producers with immunomodulatory capability[24–26]. Detailed microbiome data including overall, and two-group comparisons are presented in Supplementary Table 2.

**Bacterial genes involved in SCFA synthesis from dietary fibre characterised the microbiome of NAFLD-HCC.** To confirm that NAFLD-HCC microbiome enriched species have the functional capability to produce SCFAs, we measured the abundance of candidate bacterial genes responsible for SCFA synthesis in faecal

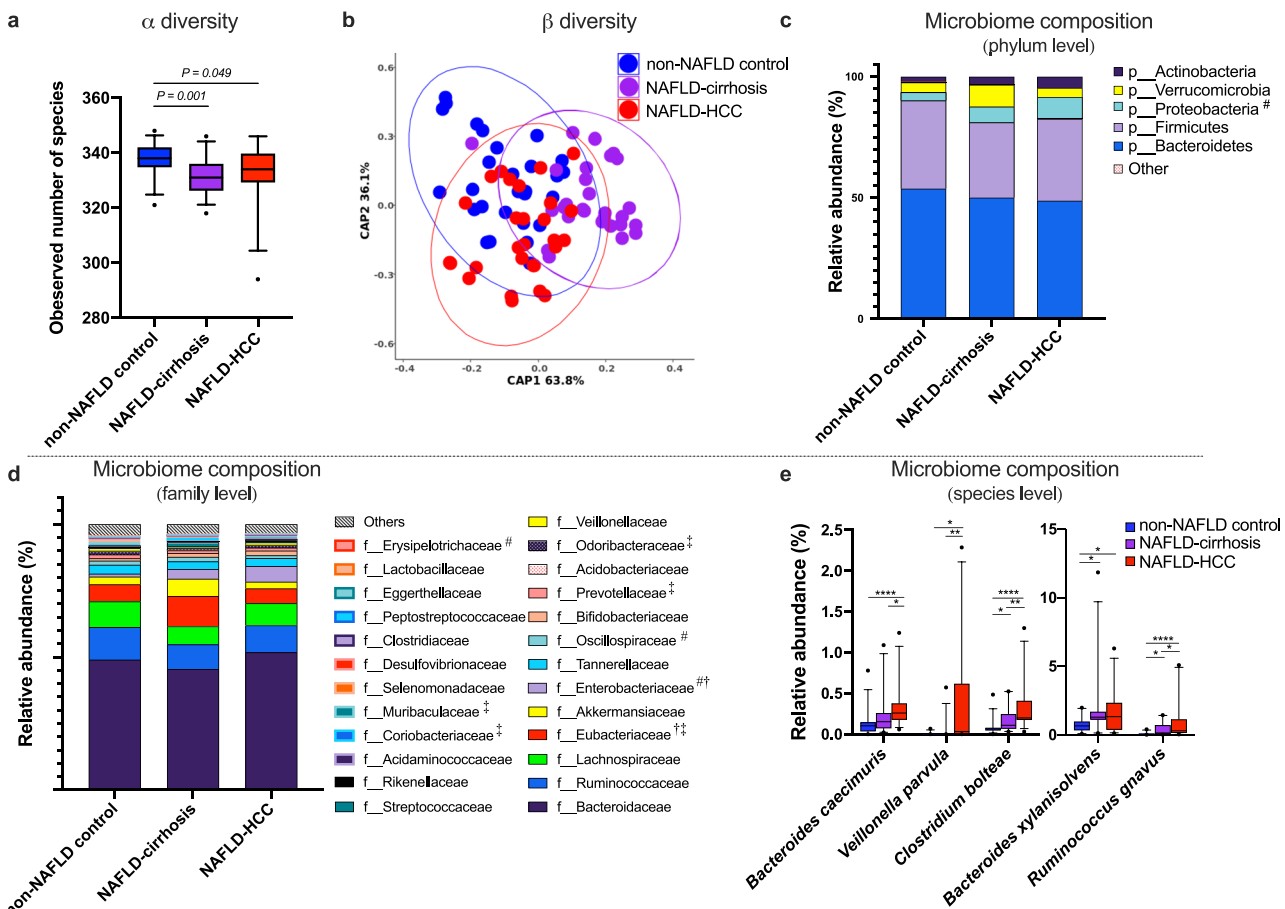

**Fig. 2 Distinct faecal microbiota profiles of subjects with NAFLD-HCC, NAFLD-cirrhosis, and non-NAFLD controls. a** Alpha-diversity based on observed number of species in non-NAFLD control, NAFLD-cirrhosis and NAFLD-HCC faecal samples. Box plots indicate median (middle line), 25th, 75th percentile (box) and 5th and 95th percentile (whiskers) as well as outliers (single points). **b** Beta-diversity using constraint analysis of principle (CAP) (controlled for age and gender) demonstrating separation of non-NAFLD control, NAFLD-cirrhosis, and NAFLD-HCC microbial communities in faecal samples at species level using 'capscale' function in R package Vegan2 ($P = 0.004$, Permutation = 999). Ellipses are added with function 'stat_ellipse' with a confidence level of 0.95. **c** Microbiome composition at phylum level in non-NAFLD control, NAFLD-cirrhosis, and NAFLD-HCC faecal samples. **d** Microbiome composition at family level showing top 25 most abundant families in non-NAFLD control, NAFLD-cirrhosis, and NAFLD-HCC faecal samples. Data in **c** and **d** are presented as mean relative abundance, with differences between groups shown as #$P < 0.05$ for NAFLD-HCC compared to non-NAFLD control, †$P < 0.05$ for NAFLD-HCC compared to NAFLD-cirrhosis and ‡$P < 0.05$ for NAFLD-cirrhosis compared to non-NAFLD control with exact $P$ values shown in Supplementary Table 2. **e** Microbiome composition at species level illustrating most enriched species in NAFLD-cirrhosis and NAFLD-HCC compared to non-NAFLD control faecal samples. Box plots indicate median (middle line), 25th, 75th percentile (box) and 5th and 95th percentile (whiskers) as well as outliers (single points). *$P < 0.05$, **$P < 0.01$, ****$P < 0.0001$ with exact $P$ shown in Supplementary Table 2. For panels **a**–**e** sample size is $n = 30$ non-NAFLD control, $n = 28$ NAFLD-cirrhosis, $n = 32$ NAFLD-HCC as biologically independent samples. For panels **a**, **c**, **d**, and **e** $P$ calculated are calculated by Kruskal–Wallis for 3 group comparison and Dunn's test for 2 group comparison. Detailed data is shown in Supplementary Table 2. Source data are provided as a Source Data file.

samples from NAFLD-HCC, NAFLD-cirrhosis, and non-NAFLD controls (Fig. 3a).

Of 21 candidate genes, shown to be important in SCFA synthesis in previous studies[27–29], the abundance of five genes were significantly different across groups (Supplementary Table 3). Importantly, *pyruvate carboxylase* (*pycA*), responsible for the production of oxaloacetate from pyruvate was overexpressed in NAFLD-HCC compared to NAFLD-cirrhosis and non-NAFLD control ($P = 0.004$) (Fig. 3b). Additionally, genes related to acetate synthesis (*phosphate acetyltransferase*; *pta*) and butyrate/acetylphosphate synthesis (*phosphate butyryltransferase*; *ptb*) were both overexpressed in NAFLD-HCC compared to NAFLD cirrhosis and non-NAFLD controls ($P = 0.020$ and $P = 0.005$, respectively) (Fig. 3b). However, not all genes important in acetate and butyrate synthesis were overexpressed in NAFLD-HCC microbiome. For instance, no difference in gene expression

across groups was seen for *acetate kinase* (*ackA*) and *butyrate kinase* (*butK*).

Furthermore, two genes in the propionate synthesis pathway, namely *fumarate reductase* (*frd*) and *succinate-CoA synthetase* (*sucC*) were overexpressed in NAFLD-HCC microbiome compared to non-NAFLD control ($P = 0.021$ and $P = 0.0006$, respectively). However, no difference in the abundance of these genes were seen between NAFLD-HCC and NAFLD-cirrhosis, nor were differences seen between NAFLD-cirrhosis and non-NAFLD control (Fig. 3b). Detailed gene abundance data including overall, and two-group comparisons are presented in Supplementary Table 3.

**Metabolomic studies revealed elevated concentrations of SCFA and their intermediates in faeces and serum from NAFLD-HCC subjects.** Faecal and serum samples from NAFLD-HCC,

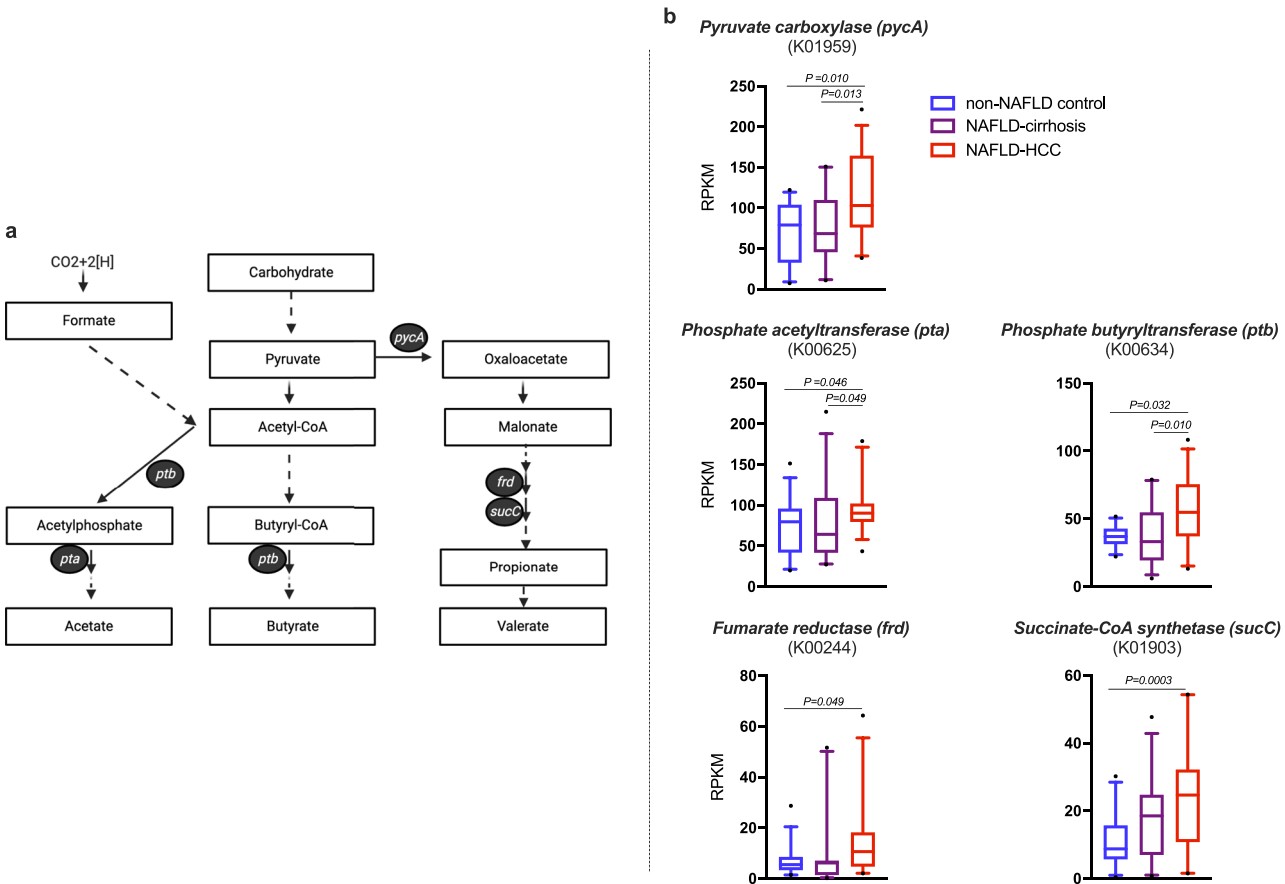

**Fig. 3 NAFLD-HCC microbiome is characterised by increased abundance of genes that mediate short chain fatty acid (SCFA) synthesis from dietary fibre. a** Pathways of bacteria-mediated SCFA synthesis. Only genes where difference in abundance was observed between groups are shown. Solid arrow represents single step pathway, whilst dashed arrow represents multistep pathway. **b** Relative abundance of SCFA synthesis genes (reads per kilobase per million reads; RPKM) in the microbiome of NAFLD-HCC compared to NAFLD-cirrhosis and non-NAFLD control. Sample size is $n = 30$ non-NAFLD control, $n = 28$ NAFLD-cirrhosis, $n = 32$ NAFLD-HCC as biologically independent samples. Box plots indicate median (middle line), 25th, 75th percentile (box) and 5th and 95th percentile (whiskers) as well as outliers (single points). $P$ calculated are calculated by Kruskal–Wallis for 3 group comparison and Dunn's test for 2 group comparison. Detailed data is shown in Supplementary Table 3. Source data are provided as a Source Data file. pycA pyruvate carboxylase, pta phosphate acetyltransferase, ptb phosphate butyryltransferase, frd fumarate reductase, sucC succinate-CoA synthetase, KO KEGG Orthology.

NAFLD-cirrhosis, and non-NAFLD control were subjected to metabolomics for measurement of SCFAs and their intermediates.

In faeces, 65 metabolites important in bacterial metabolic processes were measured with LC–MS/MS (Supplementary Table 4). Of these, three were found to be specific for NAFLD-HCC; oxaloacetate, acetylphosphate, and isocitrate. Oxaloacetate and acetylphosphate are known SCFA intermediates (Fig. 3a), and were significantly elevated in faeces of subjects with NAFLD-HCC compared to NAFLD-cirrhosis and non-NAFLD control ($P < 0.0001$ and $P = 0.0001$, respectively) (Fig. 4a), whilst levels of isocitrate was lower in NAFLD-HCC faeces compared to NAFLD-cirrhosis and non-NAFLD control ($P < 0.0001$) (Supplementary Fig. 4). Other metabolites with variable concentrations across the three groups are shown in Supplementary Fig. 4 and Supplementary Table 4.

Next, we used 1H-NMR to perform absolute quantification of six key SCFAs in both faecal and serum samples. The faeces of NAFLD-HCC subjects were enriched in acetate ($P < 0.0001$), butyrate ($P < 0.0001$), and formate ($P < 0.0001$) compared to NAFLD-cirrhosis and non-NAFLD control (Fig. 4b). Thus, increase in these faecal SCFA was NAFLD-HCC specific. There was no significant difference in levels of faecal malonate, propionate, or valerate between groups. In serum we identified increased levels of butyrate ($P = 0.005$) and propionate ($P = 0.0002$) in NAFLD-HCC

compared to NAFLD-cirrhosis and non-NAFLD control (Fig. 4c). Thus, increase in these serum SCFAs were NAFLD-HCC specific. Serum malonate was increased in both NAFLD-HCC and NAFLD-cirrhosis compared to non-NAFLD controls ($P < 0.0001$), however, no difference was seen between NAFLD-HCC and NAFLD-cirrhosis groups (Fig. 4c). There was no significant difference in levels of serum acetate, formate, and valerate between groups (Supplementary Table 4). Detailed metabolomic data including overall, and two-group comparisons are presented in Supplementary Table 4.

**Ex vivo studies showed that NAFLD-HCC microbiota elicit an immunosuppressive response.** In animal and in vitro studies, certain bacterial species and SCFAs have been shown to directly promote an immunosuppressed response[19,24]. Thus, after characterising microbiota composition and related faecal metabolomics in NAFLD-HCC subjects, we aimed to examine the effect of the microbiota on the peripheral immune response in an ex vivo cell culture model[23]. We stimulated individual peripheral blood mononuclear cell (PBMC) samples from non-NAFLD controls ($n = 10$) with individual bacterial extract (BE) from each NAFLD-HCC ($n = 10$), NAFLD-cirrhosis ($n = 10$), and non-NAFLD controls ($n = 10$) subject under various stimulation

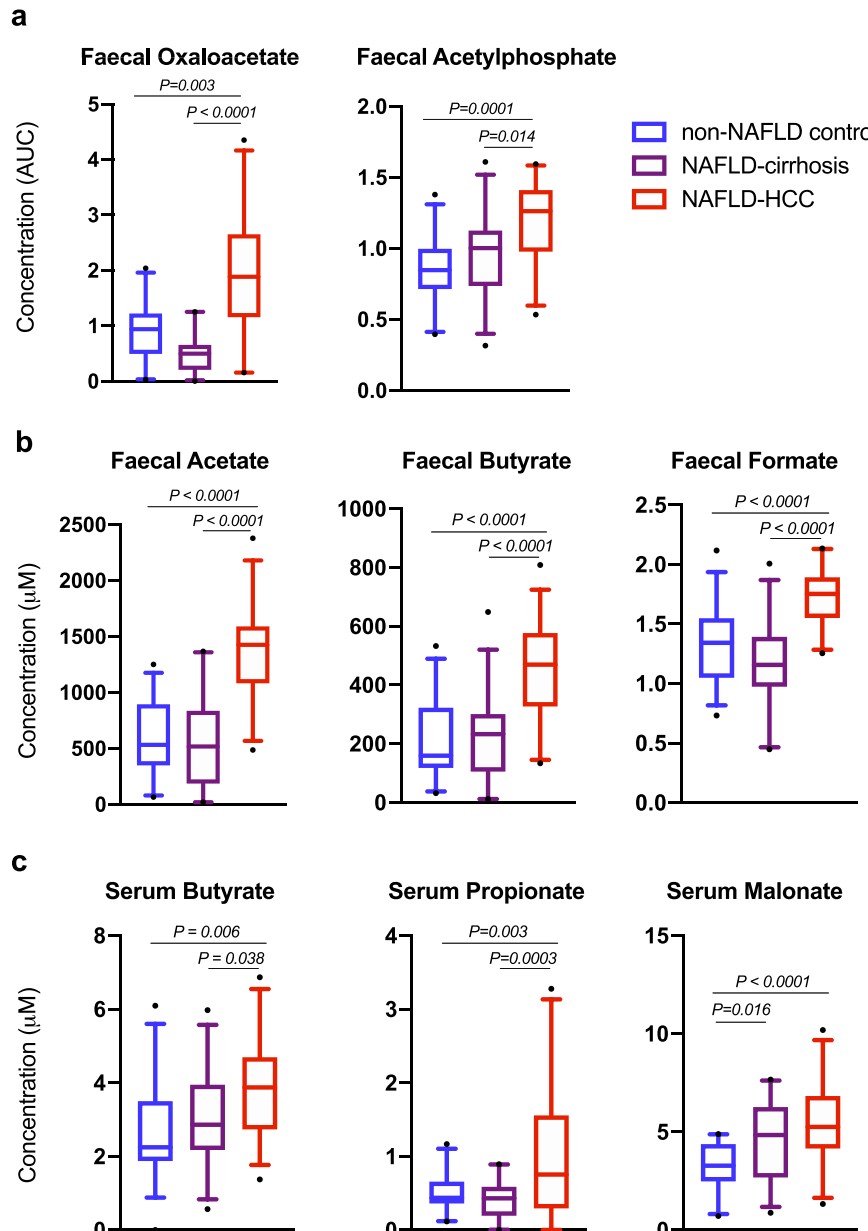

**Fig. 4 Faeces and serum from subjects with NAFLD-HCC are characterised by increased levels of some short chain fatty acids (SCFA) and SCFA-intermediates compared to NAFLD-cirrhosis and non-NAFLD controls. a** Relative quantification of SCFAs intermediates in faecal samples from non-NAFLD control, NAFLD-cirrhosis, and NAFLD-HCC subjects. Data represented as area under the curve (AUC) relative to pooled sample. **b** Absolute concentration of SCFAs in faecal samples from non-NAFLD control, NAFLD-cirrhosis, and NAFLD-HCC subjects. **c** Absolute quantification of SCFAs in serum samples from non-NAFLD control, NAFLD-cirrhosis, and NAFLD-HCC subjects. For panels **a–c** sample size is $n = 30$ non-NAFLD control, $n = 28$ NAFLD-cirrhosis, $n = 32$ NAFLD-HCC as biologically independent samples. Box plots indicate median (middle line), 25th, 75th percentile (box) and 5th and 95th percentile (whiskers) as well as outliers (single points). *P* values are calculated by one-way ANOVA for 3 group comparison and Tukey's test for 2 group comparison. Detailed data is shown in Supplementary Table 4. Source data are provided as a Source Data file.

conditions. These BE were prepared from the least diverse faecal samples, based on α-diversity in NAFLD-HCC ($321.8 \pm 11.53$; mean ± standard deviation of number of observed species) and NAFLD-cirrhosis ($325.2 \pm 2.74$) and most diverse faecal samples in non-NAFLD control group ($343.8 \pm 2.10$). Our model generated a total of 800 samples and 7270 flow cytometer readings.

**NAFLD-HCC, but not NAFLD-cirrhosis microbiota induced the expansion of effector IL-10+ Tregs, whilst attenuating the expansion of cytotoxic CD8+ T cells.** The BE from NAFLD-HCC subjects induced expansion of Tregs beyond that seen in NAFLD-cirrhosis and non-NAFLD control BE ($P = 0.002$) (Fig. 5a). We determined the phenotype of this expanded Treg population and confirmed that BE from NAFLD-HCC subjects induced the expansion of effector IL-10+ Tregs (CD3+CD4+CD25+Foxp3+IL-10+) compared to non-NAFLD control BE ($P = 0.011$) (Supplementary Fig. 5a). In contrast, BE from NAFLD-HCC subjects attenuated the expansion of total CD8+ T cells (CD3+CD8+) compared to both NAFLD-cirrhosis and non-NAFLD control BE ($P = 0.001$) (Fig. 5b). We confirmed that this observation was not a result of apoptosis (Supplementary

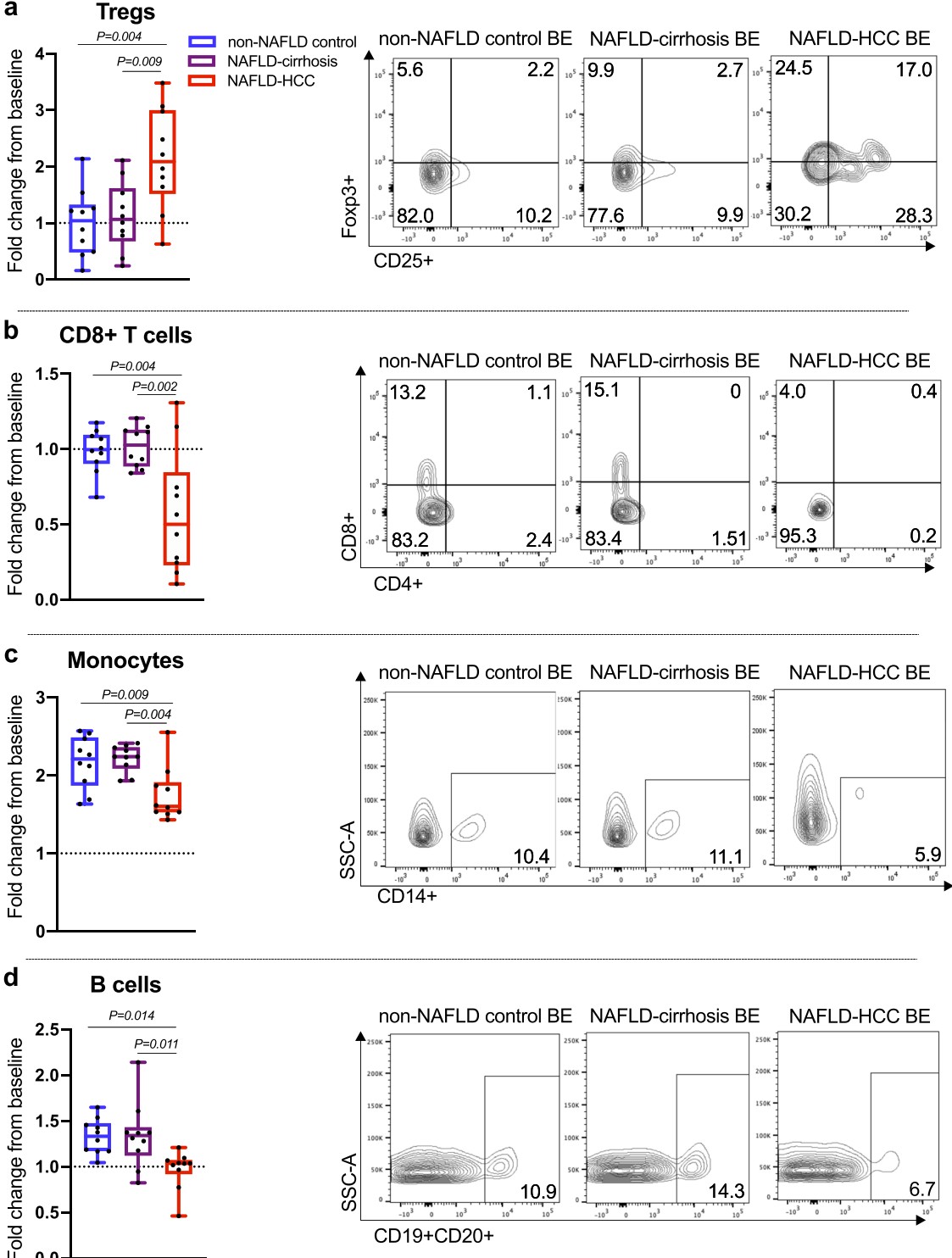

**Fig. 5 Bacterial extract (BE) from subjects with NAFLD-HCC elicit an immunosuppressive phenotype in PBMCs from non-NAFLD subjects.** Fold change from baseline and representative flow cytometry plots of **a** Tregs (CD3+CD4+CD25+Foxp3+) in response to non-NAFLD control, NAFLD-cirrhosis, and NAFLD-HCC BE. **b** CD8+ T cells (CD3+CD8+) in response to non-NAFLD control, NAFLD-cirrhosis, and NAFLD-HCC BE. **c** Monocyte antigen-presenting cells (monocytes; CD3−CD14+) in response to non-NAFLD control, NAFLD-cirrhosis, and NAFLD-HCC BE and **d** B cells (CD19+CD20+) in response to non-NAFLD control, NAFLD-cirrhosis, and NAFLD-HCC BE. Data is represented as % of viable CD3+ lymphocytes for Tregs and CD8+ T cells, and % of viable singlets for monocytes and B cells, normalised to baseline measurements; baseline immune response is represented as dashed line. For panels **a–d** sample size is $n = 10$ non-NAFLD control, $n = 10$ NAFLD-cirrhosis, $n = 10$ NAFLD-HCC as biologically independent samples. Box plots indicate median (middle line), 25th, 75th percentile (box) and 5th and 95th percentile (whiskers) as well as outliers (single points). $P$ values are calculated by one-way ANOVA for 3 group comparison and Tukey's test for 2 group comparison. Source data are provided as a Source Data file.

Fig. 5d). Furthermore, we observed that BE from NAFLD-HCC subjects attenuated the expansion of cytotoxic CD8+ T cells (CD3+CD8+CCR7−CD45RO−) and promoted the emergence of central memory CD8+ T cells (CD3+CD8+CCR7+CD45RO+) (P < 0.0001 and P = 0.003, respectively) (Supplementary Fig. 5b).

Similarly, BE from NAFLD-HCC subjects attenuated the expansion of total CD4+ T cells (CD3+CD4+) compared to NAFLD-cirrhosis and non-NAFLD control BE (P < 0.0001) (Supplementary Fig. 5c). No significant change was seen in T helper (Th1; CD3+CD4+Tbet+) populations (Supplementary Fig. 5c). We again confirmed that apoptosis was not contributing to CD4+ T cell attenuation induced by NAFLD-HCC BE (Supplementary Fig. 5d). Detailed immune cell population data including overall, and two-group comparisons are presented in Supplementary Table 1.

**NAFLD-HCC microbiota altered antigen presenting milieu.** To further characterise changes in the peripheral immune response, we measured antigen presenting cell populations in our ex vivo model including monocytes (CD3−CD14+), myeloid dendritic cells (CD3−HLADR+CD11c+) and B-cells (CD19+CD20+). The BE from NAFLD-HCC subjects attenuated the expansion of monocytes compared to NAFLD-cirrhosis and non-NAFLD control BE (P = 0.002) (Fig. 5c). Similarly, BE from NAFLD-HCC subjects attenuated the expansion of B-cells compared to NAFLD-cirrhosis and non-NAFLD control BE (P = 0.006) (Fig. 5d). Detailed immune cell population data including overall, and two-group comparisons are presented in Supplementary Table 1.

**NAFLD-HCC microbiota elicited a cytokine milieu that supports immunosuppression.** Studies have shown IL-2 and IL-12 (p70) to be important in CD8+ T cell expansion and activation[30–32]. We observed that BE from NAFLD-HCC subjects dampened the production of these pro-inflammatory cytokines compared to both NAFLD-cirrhosis and non-NAFLD control BE (P = 0.023 and P = 0.001, respectively) (Fig. 6). Additionally, both NAFLD-HCC and NAFLD-cirrhosis BE dampened the production of IL-4 compared to non-NAFLD control BE (P < 0.0001) (Fig. 6). A corresponding increase in IL-6 and the anti-inflammatory cytokine, IL-10 was induced by NAFLD-HCC BE compared to both NAFLD-cirrhosis and non-NAFLD control BE (P = 0.001 and P = 0.003, respectively) (Fig. 6). Detailed cytokine data including overall, and two-group comparisons are presented in Supplementary Table 5.

**Correlative models confirmed a link between NAFLD-HCC microbiota, butyrate, and an immunosuppressed response.** We performed correlation analysis to assess the relationship between the abundance of NAFLD-HCC-enriched species, concentration of faecal metabolites and T cell responses elicited in our ex vivo model. Indeed, significant correlations were consistent with our observed experimental results. Specifically, B. caecimuris, B. xylanisolvens, and C. bolteae were positively correlated with effector IL-10+ Tregs (R = 0.9, P = 0.010, R = 0.66, P = 0.021 and R = 0.63, P = 0.031, respectively). Importantly, B. caecimuris and B. xylanisolvens were also negatively correlated with CD8+ T cells (R = −0.56, P = 0.021, R = −0.72, P = 0.032), highlighting the importance of these taxa in modulation of adaptive immunity. Additionally, V. parvula was positively correlated with Tregs (R = 0.65, P = 0.033), but not effector IL-10+ Tregs, and negatively correlated with CD8+ T cells (R = −0.81, P = 0.024), whereas, R. gnavus was negatively correlated with cytotoxic

CD8+ T cells (R = −0.64, P = 0.012) but not with Tregs or effector IL-10+ Tregs (Fig. 7a).

With respect to SCFAs and their intermediates, a positive correlation was seen between butyrate and Tregs (R = 0.67, P = 0.013) and effector IL-10+ Tregs (R = 0.69, P = 0.031), a finding that supports previously reported in vitro studies that have shown butyrate to induce Treg populations in PBMCs[20]. Additionally, negative correlations were seen between butyrate and cytotoxic CD8+ T cells (R = −0.80, P = 0.024), again highlighting the importance of this SCFA in modulation of the peripheral immune response. No significant positive or negative correlations were seen between acetylphosphate, oxaloacetate, acetate or formate and Tregs or CD8+ T cells (Fig. 7b). Correlations between all measured metabolites and T cell responses elicited in our elicited ex vivo are shown in Supplementary Fig. 6.

## Discussion

The gut microbiota and its metabolites have been proposed as cofactors in liver disease progression and the development of HCC through their interaction with immune compartments via the gut–liver axis. In this study, we employed shotgun metagenomics and metabolomics analysis to identify structural and functional changes of the microbiome that discriminate NAFLD-HCC from NAFLD-cirrhosis. Additionally, we utilised an ex vivo cell culture model to demonstrate a NAFLD-HCC-specific effect of this microbiota and its metabolites on the peripheral immune that have important clinical implications.

The presented data add to several lines of evidence that gut dysbiosis characterises liver cirrhosis[5–7,33,34]. Despite the use of different control groups, and geographical variations, human metagenomic data from our cohort and others[6,7], support an emerging core microbiome signature that characterises NAFLD-cirrhosis, with enrichment of R. gnavus, C. bolteae, Streptococcus parasanguinis, and Klebsiella pneumoniae, and reduced abundance of beneficial species including Faecalibacterium prausnitzii, Alistipes putredinis, and Eubacterium eligens. We further identified V. parvula and B. caecimuris to distinguish NAFLD-HCC from NAFLD-cirrhosis[6]. Similarly, we observed some similarities in the detected metabolomic profile of NAFLD-cirrhosis, in our study and others[6], whereby microbial metabolism of tryptophan to indole 3-carboxylate were increased in NAFLD cirrhosis[6]. Therefore, emerging data support the potential for faecal microbial and metabolomic signatures to act as a global diagnostic tool for NAFLD-cirrhosis.

As for HCC-specific microbial signatures, promising data are emerging[8,9]. In agreement with our findings, 16S rRNA analyses of patients with NAFLD-HCC have detected enrichment in Bacteroides and Ruminococcaceae members, which correlated with several of the systemic inflammatory and immune markers[8]. Ren et al.[9] using 16S rRNA analyses of faecal specimens from patients with chronic hepatitis B-related HCC were able to detect HCC-specific microbial markers, which were successfully cross validated in other Chinese cohorts. In contrast to our findings and others[6], Ren et al. reported in patients with early HCC, a decrease in butyrate-producing bacterial families namely Ruminococcus, Oscillibacter, Faecalibacterium, Clostridium IV, and Coprococcus[9]. These disparate findings are plausibly due to significant differences in the study cohorts. In the study by Ren et al. chronic hepatitis B infection was the aetiology of the underlying liver disease in the studied cohort who additionally had a much lower BMI ($22.8 \pm 2.04 \, \text{kg/m}^2$), compared to our study cohort with a high BMI ($31.5 \pm 7.6 \, \text{kg/m}^2$) and many metabolic risk factors. Clearly, sufficiently powered studies in large cohorts are still needed to determine liver disease specific alterations of the microbiome with HCC.

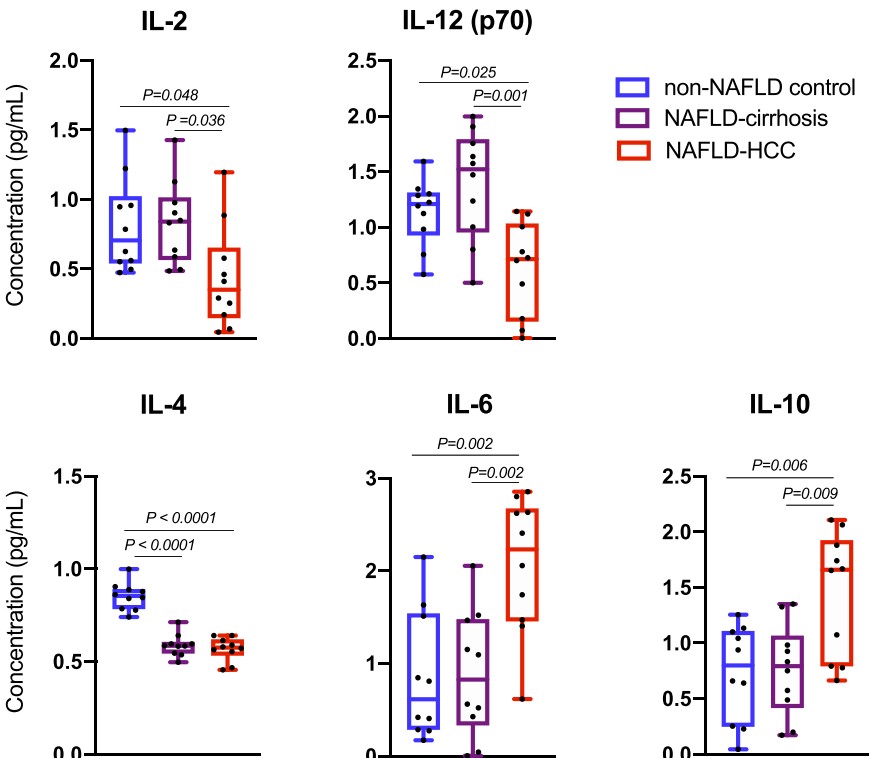

**Fig. 6 Bacterial extract (BE) from subjects with NAFLD-HCC alter pro-/anti-inflammatory cytokine milieu.** Quantification of cytokines in PBMC culture media following addition of non-NAFLD control, NAFLD-cirrhosis, and NAFLD-HCC bacterial extract (BE). Sample size is $n = 10$ non-NAFLD control, $n = 10$ NAFLD-cirrhosis, $n = 10$ NAFLD-HCC as biologically independent samples. Box plots indicate median (middle line), 25th, 75th percentile (box) and 5th and 95th percentile (whiskers) as well as outliers (single points). $P$ values are calculated by one-way ANOVA for 3 group comparison and Tukey's test for 2 group comparison. Detailed data is shown in Supplementary Table 5. Source data are provided as a Source Data file.

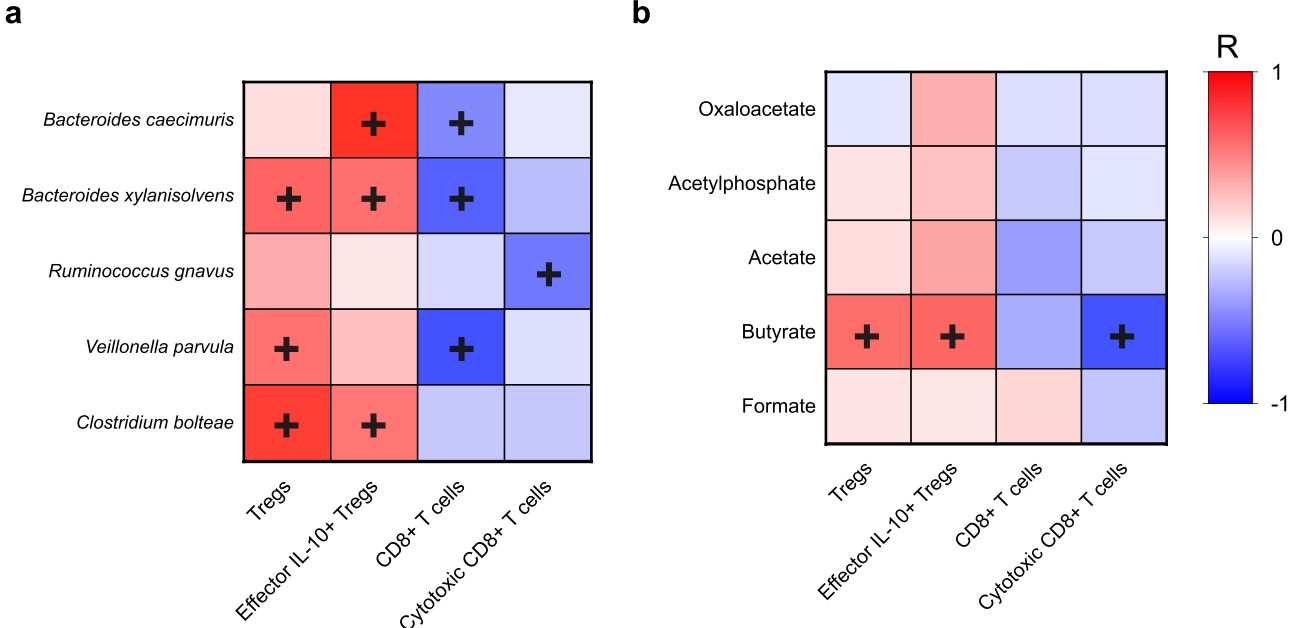

**Fig. 7 Enriched species in the microbiome of subjects with NAFLD-HCC and the metabolite butyrate, correlate with the elicited immune response ex vivo.** Heatmap of Spearman's rank correlation of (**a**) enriched species in microbiome of subjects with NAFLD-HCC and T cell immune responses measured ex vivo (**b**) enriched metabolites in faeces of subjects with NAFLD-HCC and T cell immune responses measured ex vivo. Colour legend represents correlation coefficient ($R$). "+" denotes significant correlation ($P < 0.05$) are shown after Benjamini–Hochberg correction. All data is shown in Supplementary Fig. 6. Source data are provided as a Source Data file.

We further observed the microbiome in NAFLD-HCC to be enriched in genes required for SCFA synthesis from dietary fibre including *pyruvate carboxylase*, *phosphate acetyltransferase*, and *phosphate butryltransferase*. Concentrations of SCFA (particularly butyrate) and their intermediates in the faeces and serum of subjects with NAFLD-HCC exceeded that seen in NAFLD-cirrhosis, suggesting that microbial species in NAFLD-HCC act synergistically to increase SCFA production. Indeed, increased levels of faecal SCFA have been reported in obese patients with NAFLD[35], however, our study cohort was matched for BMI, hence obesity per se does not account for the increased SCFA concentrations seen specifically in NAFLD-HCC, and more likely attributable to the composition of the microbiota in patients with NAFLD-HCC. In support of our findings, recent animal studies have demonstrated that in the context of gut dysbiosis, mice fed high fibre diet exhibited increased SCFA production (particularly butyrate) which promoted hepatocyte proliferation, liver fibrosis, and hence HCC[36]. Further, treatment of animal models of melanoma with vancomycin has been observed to improve responses to radiotherapy and reduce tumour burden by reducing SCFA concentrations through abrogating *Ruminococcaceae* and *Lachnospiraceae* members[22]. These studies speculated that generation of large amounts of SCFA, particularly butyrate in a context of dysbiosis, may instead create a tumour-promoting microenvironment[36]. However, contrary to the inflammation and tumour-promoting potential of SCFA, other investigators have found that mice fed a high-fibre diet had increased circulating levels of SCFA, and were protected against inflammation, and cancer[37,38]. This SCFA-related functional paradox supports the notion that the function of SCFA at the cellular level is contextually dependent on the cell-type, and the amount of exposure. Along this line, emerging literature supports the concept that increased doses of SCFA, that exceed the threshold tolerable by the host, are more likely to promote tissue inflammation[39,40] and tumour-promoting events that outweigh any of its beneficial effects[36,41,42].

In the presented ex vivo model, the gut microbiota in NAFLD-HCC promoted expansion of total and effector IL-10+ Tregs, with reduced expansion of CD8+ T cells. Some of the identified consortium of microbial species in NAFLD-HCC, have been previously shown to play a key role in regulating the expression of Tregs. In this regard, in animal models of colitis, several *Clostridiales* strains, and *R. gnavus* have resulted in skewing the immune response toward a Treg response[24–26]. Further, colonisation of germ free mice with human microbiota has identified a mixture of Clostridia strains that are capable of inducing Tregs as well as IL-10[25,43]. Tregs promote immunotolerance, with several studies identifying Treg-mediated attenuation of cytotoxic CD8+ T cell function as being critical in this process[17]. IL-2 is required for CD8+ T cell activation as well as expansion, and Treg consumption of IL-2 is proposed as the mechanism by which Tregs attenuate CD8+ T cells in peripheral blood[30,31]. This reciprocal T cell expression profile was elicited ex vivo by the microbiota from patients with NAFLD-HCC, but not cirrhosis, suggesting an immunomodulatory effect specific for the NAFLD-HCC microbiota.

Added to the above data, our correlative models have also confirmed a link between the metabolite, butyrate, enriched in NAFLD-HCC stool, and the elicited T cell responses. Previous ex vivo studies have demonstrated that incubation of SCFAs, particularly butyrate with PBMCs result in expansion of Tregs in a concentration-dependant manner[20,21], through mechanisms that enhance Foxp3 expression[38,44,45]. Our combined ex vivo and correlative models demonstrate the *collective effect* of dysbiosis (microbiome and related metabolites) in NAFLD-HCC on T cell responses.

We also observed butyrate to negatively correlate with cytotoxic CD8+ T cells, in agreement with animal data showing butyrate to directly attenuate antigen-specific CD8+ T cell responses[22]. Contrastingly, other investigators have shown butyrate to enhance CD8+ T cell effector function[46]. Again, this suggests that the function of butyrate is dependent on its dose and specific cellular environment[39,40,47].

The ability of the microbiota and its metabolites to modulate the immune response is relevant to the HCC theme. Increased circulating Tregs in the peripheral blood of patients with HCC has been shown to independently predict a poor prognosis[17,48]. Similarly, reduction in the levels of circulating Tregs in HCC patients receiving systemic therapy, was associated with better treatment and disease outcomes due to reinvigoration of anti-tumor CD8+ T cell responses[49]. Recent studies have confirmed the importance of the gut microbiota in either promoting or inhibiting anticancer CD8+ T cell responses[50]. Collectively, it can be postulated, based on our findings and others, that gut-derived interventional strategies could be of therapeutic benefit in patients with NAFLD-HCC to reinvigorate CD8+ T cell anti-tumour responses.

A question now arises as to how gut microbiome, and/or related metabolites from NAFLD-HCC, can influence the peripheral immune response. Translocation of bacterial products and metabolites is thought to occur in cirrhosis due to intestinal barrier dysfunction[51]. To this effect, elevated levels of bacterial lipopolysaccharide (LPS) and markers of intestinal barrier dysfunction (zonulin-1) have been shown to be elevated in peripheral blood of patients with NAFLD-cirrhosis, with and without HCC[8]. Furthermore, studies have shown that the gut microbiota can *instruct* the peripheral immune response toward immunotolerance[52]. In this regard, the generation of peripheral antigen-specific populations of Tregs in response to commensal microbiota has been demonstrated in animal models[52]. Similarly, *sensing* of microbial metabolites (particularly SCFAs) by immune cells in the periphery has been demonstrated to affect the balance between pro-inflammatory and anti-inflammatory immune cells. Findings from the presented ex vivo model provide evidence that the microbiota and its metabolites in NAFLD-HCC can direct the peripheral immune response towards an immunosuppressive phenotype that is associated with worse outcomes as previously shown[17,48].

There are some limitations to our study. Firstly, for the microbiome signatures, patients with metabolic syndrome, but without NAFLD, or NAFLD without cirrhosis, could plausibly have made an appropriate control group. As a result, the current analysis cannot determine the potential confounding impact of metabolic variables on the detected microbial signatures. However, in clinical practice, it would be challenging to recruit a modest size control group matched in age to our study cohort, with metabolic syndrome, yet without NAFLD. As it is, the presence of metabolic syndrome, in particular diabetes in this population substantially increases the prevalence of NAFLD, NAFLD fibrosis, and cirrhosis[53]. Secondly, non-targeted metabolomic analysis could have provided additional valuable information. Finally, detailed information pertaining to diet composition would be relevant as SCFAs was particularly increased in the NAFLD-HCC group.

Taken together, we have identified key species enriched in NAFLD-HCC with functional and metabolomic evidence of increased SCFA synthesis, that result in an immunosuppressed response, not seen with NAFLD-cirrhosis. Our data support the notion that harnessing knowledge of the microbiome and its

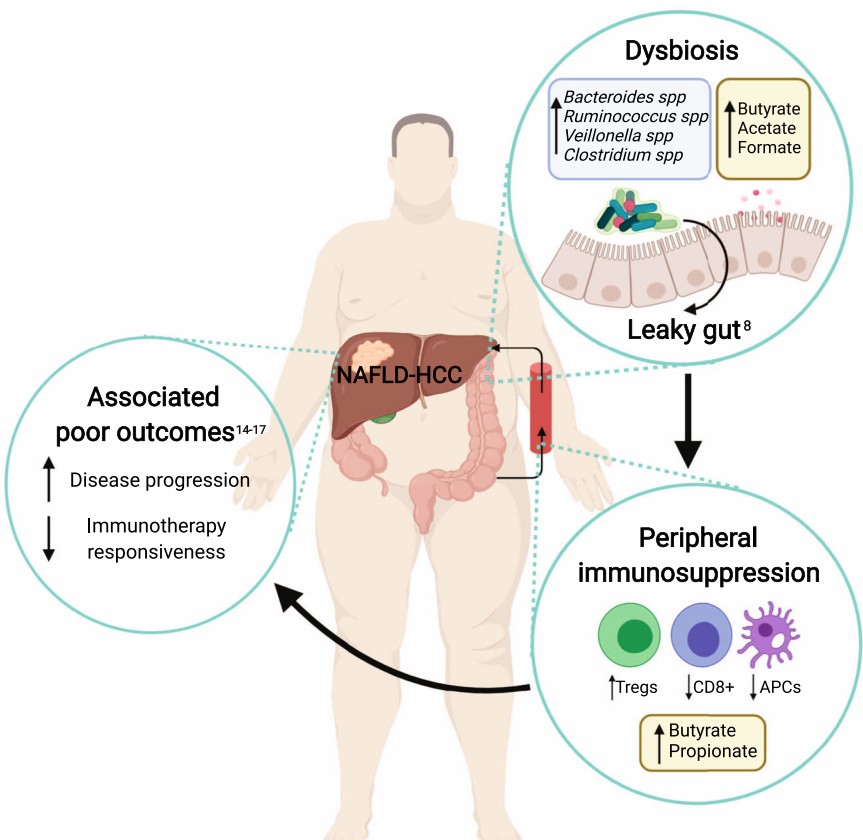

**Fig. 8 The microbiome and metabolome of subjects with NAFLD-HCC elicit an immunosuppressive phenotype ex vivo.** Schema summarising key findings. Dysbiosis in NAFLD-HCC is characterised by increased abundance of distinct bacterial species, increased functional capacity for the production of SCFAs and increased faecal SCFA concentrations. 'Leaky gut' is well described in cirrhosis with and without HCC. The microbiome and metabolome of subjects with NAFLD-HCC elicits an immunosuppressive phenotype ex vivo (characterised by increased Tregs, reduced CD8+ T cells, and reduced APCs) that has been associated with poor clinical outcomes and immunotherapy resistance in patients with HCC. Created with Biorender.com.

metabolites in HCC may be of benefit in developing prognostic markers and new therapeutic strategies (Fig. 8).

## Methods

**Study population**. Three groups were consecutively recruited to the study. These were: (1) subjects with NAFLD cirrhosis-related HCC (NAFLD-HCC) who were undergoing surgical resection; (2) Subjects with NAFLD-related liver cirrhosis (NAFLD-cirrhosis), and (3) non-NAFLD controls. All patients were matched for age, gender, and body mass index (BMI). Patients with liver disease were matched for extent of liver disease based on Child-Pugh and Model for End-Stage Liver Disease-Sodium (MELD-Na) scores. NAFLD was diagnosed based on the American Association for the Study of Liver Diseases practice guidelines[54]. Liver cirrhosis was confirmed based on clinical, biochemical, transient elastography, and radiological assessment in the NAFLD-cirrhosis group; and additionally, confirmed on histology in resection specimens in the NAFLD-HCC group. HCC was diagnosed according to international guidelines, integrating history, physical examination, biochemistry, and imaging techniques obtained by multiphasic CT, and/or dynamic contrast-enhanced MRI[55]. Diagnosis of HCC was further confirmed by histopathological examination of surgical resection specimens. Treatment decisions for HCC were determined in multidisciplinary meetings, based on international guidelines[55]. As per guidelines, surgical resection was undertaken for early HCC, when hepatic function is preserved, and sufficient remnant liver volume is maintained[55]. For the groups with liver cirrhosis, exclusion criteria included: subjects aged <18 years; alcohol consumption ≥30 g daily for men and ≥20 g daily for women, other causes of liver cirrhosis (including viral, alcoholic, autoimmune, cholestatic liver diseases, inherited liver diseases, etc.); previous clinical or biochemical evidence of hepatic decompensation; any degree of portal hypertension (based on clinical, radiological, and endoscopy or pre-operative hepatic venous pressure gradient (HVPG)); other primary liver cancers (e.g. mixed HCC and cholangiocarcinoma); probiotic or antibiotic administration (per oral or intravenous) in the 3 months prior to recruitment; known gastrointestinal disease; previous gastrointestinal surgery; regular proton-pump inhibitor or lactulose therapy. For non-NAFLD controls, a detailed medical history was taken including medications that could lead to steatosis. All participants underwent a physical

examination and had routine blood tests (including liver function tests, serological tests for hepatitis B surface antigen, hepatitis C virus antibody, autoimmune screen, and iron studies) in addition to a screening liver ultrasound to exclude hepatic steatosis. Additional exclusion criteria for non-NAFLD controls included subjects aged <18 years; alcohol consumption ≥30 g daily for men and ≥20 g daily for women, known history of liver disease (including viral, alcoholic, autoimmune, cholestatic liver diseases, and inherited liver diseases), liver ultrasound demonstrating hepatic steatosis, known gastrointestinal disease, previous gastrointestinal surgery, regular proton-pump inhibitor, lactulose therapy, or antibiotic/probiotic administration (per oral or intravenous) in the 3 months prior to recruitment.

A blood sample was taken from all subjects for determination of liver function parameters, platelet count, alpha-fetoprotein, serum SCFA levels and peripheral blood mononuclear cell (PBMC) isolation. Blood and faecal samples were collected from all subjects at the time of recruitment, and in the week preceding surgery for those in the NAFLD-HCC group. Subjects with liver disease underwent transient elastography measurement with FibroScan® at the time of recruitment. All study participants were requested to provide faecal samples in a sterile solution free tube and in DNA stabiliser (Stratech, Invitek Molecular #1038111200-STM) which were kept at 4 °C and returned to study investigators within 24 h of recruitment.

The study was approved by Sydney Local Health District Human Research Ethics Committee (HREC), New South Wales Health: approval number HREC/16/RPAH/701; SSA18/G/058. Informed consent was obtained from all study participants.

**DNA extraction and shotgun metagenomic sequencing**. DNA was extracted from faecal samples using PSP® Spin Stool DNA Kit (ThermoFisher, #IVK1038110300) as per manufacturer's instructions. Faecal samples were agitated and incubated in Thermoshaker (LabGear, Grant Thermoshaker #GRANPHMT-PSC24) and mechanically lysed using Qiagen TissueLyser II (Qiagen, #85300). Following removal of impurities, DNA was eluted, and concentration measured with Qubit dsDNA HS Assay Kit (Life Technologies, #Q32853) and Qubit Fluorometer (Life Technologies). Purified DNA was aliquoted for downstream applications and stored at −80 °C. For shotgun metagenomic sequencing, library preparation was performed with Nextra DNA Flex Library Prep (Illumina,

#20018705) for Nextra DNA CD Indexes (Illumina, #20018708) as per manufacturer's instructions. Libraries were sequenced on NovaSeq 6000 Sequencing System (Illumina, #20012850) at the Ramaciotti Centre for Genomics (UNSW Sydney).

**Measurement of metabolites in faecal samples.** To measure metabolites in faecal samples, 50 mg of fresh faeces was suspended in 1.5 mL of PBS and passed through a 40 μm cell strainer. 100 mL of sample was collected following centrifugation at 7200×$g$ for 10 min. Targeted metabolomic analysis was performed on a liquid chromatography–tandem mass spectrometry (LC–MS/MS) system consisting of an Agilent 1260 Infinity LC system (Santa Clara, CA, USA) coupled to a QTRAP 5500 mass spectrometer (AB Sciex, Foster City, CA, USA). Hydrophilic interaction chromatography, with an Amide XBridge HPLC column (Waters, Millford, MA; #186004860; 4.6 mm × 100 mm, 3.5 μm) was used to separate and analyse 65 polar metabolites, including amino acids; nucleotide and nucleoside phosphates; high-energy intermediates; organic acids; Krebs cycle intermediates and glycolytic intermediates[56]. Mobile Phase A consisted of HPLC grade acetonitrile:water (5:95% v/v) (Fisher Chemical, #FSBW6-4) with 20 mM ammonium acetate (Sigma-Aldrich, #73594-25G-F) and 20 mM ammonium hydroxide (Fluka, #44273), while Mobile Phase B consisted of 100% acetonitrile (Fisher Chemical, #A955-4). A full-scan analysis over 70–800$m/z$ at 70,000 resolution and a 3-Hz data acquisition rate was used.

**Absolute quantification of SCFAs in faecal and serum samples.** Analytical standards for SCFAs of interest were obtained in Organic Acid Kit (Sigma-Aldrich, #47264). Faecal and serum samples for $^1$H NMR were prepared by adding 350 mL deuterium oxide (D$_2$O) to 250 μL of serum or 50 mg of fresh faeces. Samples were centrifuged at 14,000×$g$, for 5 min at 4 °C. 250 mL of supernatant was passed through Amicon Ultra-0.5 Centrifugal Filter Units (3 kDa, 0.5 mL) (Merck, #UFC500396) and centrifuged at 14,000×$g$, for 20 min at 40 °C. The filtrate was incubated with deuterium chloroform and deuterium methanol on ice for 10 min, then centrifuged for 10 min at 14,000×$g$. The polar phase was added to 60 mL of 200 mM trisodium phosphate/D$_2$O and 18 mL of 5 mM 4,4-dimethyl-4-silapentane-1-sulfonic acid (DSS)/D$_2$O to a final volume of 180 mL. $^1$H NMR spectra were acquired at 298 K using the Bruker lc1pncwps pulse sequence (total of 96 scans), with irradiation of water resonance applied during pre-saturation delay and mixing time. $^{13}$C continuous wave decoupling was applied during acquisition. Data were processed and analysed using the Chenomx® NMR Suite Professional (Chenomx Inc., Edmonton, AB, Canada). All samples were referenced to a 0.5 mM DSS and the Human Metabolome Database (www.hmdb.ca).

**Bacterial extract preparation for stimulation of human PBMCs.** Total bacterial extracts (BE) were prepared from faecal samples obtained from subjects with NAFLD-HCC, NAFLD-cirrhosis, and non-NAFLD controls by suspending 50 mg of fresh faeces in 1.5 mL of PBS, followed by filtration through a 40 μm cell strainer. Samples were then resuspended in 1.5 mL of PBS supplemented with addition of protease inhibitor (Roche, #4693159001) and phosphatase inhibitor (Roche, #4906845001). Samples were then heat-inactivated at 65 °C for 1 h and sonicated for 10 min. Protein concentration in the resulting suspension was measured using the Pierce BCA protein assay kit (Thermo Scientific, #23227) and LPS levels were measured with Pierce Chromogenic Endotoxin Quantification Kit (ThermoFisher, #A39552) as per manufacturer's instructions.

**Human PBMC isolation and stimulation with bacterial extract.** Peripheral blood mononuclear cells (PBMCs) were isolated from non-NAFLD control participants. Following blood sample collection, 10 mL of whole blood was centrifuged at 700×$g$ for 10 min. Plasma was removed, and PBS added to volume of 30 mL. Next, 15 mL of Ficoll® solution (Merck, #26873-85-8) was gently underlayed with a mixing cannula, and centrifuged at 700×$g$ for 25 min at room temperature. PBMCs were harvested from the Ficoll® separation layer; cell count and viability were assessed with light microscopy using 20 μL of PBMC sample mixed with 20 μL of 0.4% trypan blue solution (ThermoFisher, #15250061). PBMCs were resuspended in 10% DMSO in RPMI media supplemented with autologous plasma and stored in liquid nitrogen for downstream experiments.

An experimental approach based on previously published studies was used to stimulate PBMCs with BE under several differentiation conditions (Treg, Th1, and undifferentiated)[23]. In detail, PBMCs were seeded in flat-bottom culture plates (10$^6$ cells/mL) and incubated with RPMI media supplemented with 10% FBS (ThermoFisher, #10099141) and 1% penicillin/streptomycin (Life technologies, #15140122) and 1% glutamine (Life Technologies, #35050061). Prior to PBMC stimulation, all BE were prepared to protein concentration of 10 mg/mL, as higher concentrations were shown in dose-titration experiments to compromise PBMC viability to <90% (Supplementary Fig. 7). BE concentrations were subsequently normalised to LPS levels. Each subject's individual BE was used to stimulate each PBMC sample (i.e. self and non-self BE stimulation). For human Treg differentiation, PBMCs were stimulated with anti-human CD3 (0.3 μg/mL; BD Biosciences, #555336), anti-human CD28 (2 μg/mL; BD Biosciences, #555725), and recombinant human TGF-β1 (2.5 ng/mL; R&D Systems #240B002) for 3 days in the presence of BE. For human Th1 differentiation, PBMCs were stimulated with

anti-human CD3 (2 μg/mL; BD Biosciences #555336), anti-human CD28 (2 μg/mL; BD Biosciences, #555725), anti-human IL-4 (5 μg/mL; BD Biosciences #554605), and recombinant human IL-12 (20 ng/mL; BD Biosciences #554613) for 3 days in the presence of BE. After 3 days, the culture media was changed and Th1 cells were restimulated for 4 h with ionomycin (1 μM; Sigma-Aldrich, #I3909) and PMA (50 ng/mL; Sigma-Aldrich, #P1585) in the presence of monensin (2 μM; Life Technologies, #00-4505-51). Undifferentiated PBMCs were stimulated for 3 days in the presence of BE. Cells were harvested and centrifuged at 300×$g$ for 5 min. The supernatant was removed, followed by resuspension in 1 mL of PBS. Cell count and viability were again assessed with light microscopy using 20 μL of PBMC sample mixed with 20 μL of 0.4% trypan blue solution (ThermoFisher, #15250061).

**Flow cytometry of human PBMCs.** Fixable viability stain (BD Biosciences, #564406) was used to identify and gate live PBMCs. BD Cytofix/Cytoperm kit (BD Biosciences, #554714) was used for intracellular cytokine staining, whilst Transcription Factor Buffer Set (BD Biosciences, #562574) was used for transcription factor staining as per manufacturer's instructions. The following antibodies were used: Annexin-BV605 (1:100; BD Biosciences, #563974), CD3-PE-Cy7 (1:100; BD Biosciences, #563423), CD4-PerCP-Cy5.5 (1:100; BD Biosciences, #560650), CD25-BV421 (1:100; BD Biosciences, #562442), Foxp3-AF488 (1:25; BD Biosciences, #560047), IL-10-PE (1:25; BD Biosciences, #559337), CD8-AP7-H7 (1:100; BD Biosciences, #560179), CD45RO-BV711 (1:100; BD Biosciences, #563722), CCR7-PE (1:25; BD Biosciences, #560765), CD14-APC (1:25; BD Biosciences, #555399), CD11c-PE (1:25; BD Biosciences, #555392), HLA-DR-APC-H7 (1:100; BD Biosciences, #561358), CD19-BB515 (1:400; BD Biosciences, #564456), CD20-BB515 (1:400; BD Biosciences, #564568), and T-bet-PE (1:100 BD Biosciences, #561265). Flow cytometry was performed on BD LSRFortessa (TM) X-20 Cell Analyzer (BD Biosciences, #657675R1). Data was collected on BD FACSDiva (TM) software (v8.0.3) (BD Biosciences) and analysed using FlowJo software (v10.5.3) (TreeStar). PBMCs were gated to identify the lymphocyte population based on forward and side scatter, followed by gating for single cell and live cell populations. Unstained, single colour, and fluorescence-minus-one control were used to identify stained populations of cells of interest. Gating strategy of target cell populations is shown in Supplementary Fig. 8a–d.

**Measurement of cytokines in culture media of stimulated PBMCs.** Bio-Plex Pro Human Cytokine Assay (Bio-Rad, #12007283) was used to measure 48 cytokines and chemokines in PBMC culture media, as per manufacturer's instructions. Samples were prepared at 1:4 dilution, and a total of 10 standards were used per cytokine to generate standard curves. The plate was read on MAGPIX (Luminex xMAP) and data were analysed with Bio-Plex Manager software (version 6.0).

**Bioinformatic and statistical analysis.** Metagenomic reads underwent preprocessing prior to taxonomic and functional annotation. In brief, deduplication of shotgun metagenomic reads was performed using BBmap/clumpify.sh (bbmap-v38.79-0: sourceforge.net/projects/bbmap/). Next, low-quality metagenomic reads were processed using fastp (v0.19.5)[57]. Sequence reads were mapped against the human genome (GRCm38.p6) using minimap2 (v2.16)[58], and human host sequences were removed. Compositional profiling was performed using KrakenUniq (v 0.5.8)[59]. Analysis of taxonomic datasets was completed in R (v3.6.1).

Alpha diversity metrics were calculated at species level with otusummary[60] followed by Kruskal–Wallis with Dunn's post hoc test to determine differences between groups. The constrain analysis of principals (CAP) was used to show difference of bacterial community using capscale function from R package vegan[61] and plotted using ggplot2[62], the function ANOVA with permutation of 999 was used to test the significance of variance. Ellipses are added with function stat_ellipse with a confidence level of 0.95. Linear discriminant analysis Effect Size (LEfSe) analysis and Kruskal–Wallis with Dunn's post hoc test was used to determine microbiota significantly different between groups. All $P$ values were corrected using Benjamini–Hochberg multiple test correction. $P$ values of <0.05 were considered statistically significant.

Function of the gut microbiome was annotated with HUMAnN2 to generate gene families, then transformed into Kyoto Encyclopaedia of Genes and Genomes (KEGG) Orthologies[63] using script humann2_regroup_table (-g uniref90_ko). Gene abundance was normalised with reads per kilobase of transcript, per million mapped reads (RPKM) for downstream analysis. Kruskal–Wallis with Dunn's post hoc test was used to determine gene abundance significantly different between groups. $P$ values of <0.05 were considered statistically significant.

Patient metadata was recorded in Microsoft Excel (v16.41). Metabolite, cytokine and PBMC analysis was performed in Prism v8.2.1 (GraphPad Software, Inc.) and one-way ANOVA with Tukey's multiple comparisons test was used to determine differences between groups. Spearman's rank correlation was performed in R (v3.6.1) to analyse relationships between the microbiota, faecal metabolites and immune changes as measured in our ex vivo model. All $P$ values were corrected using Benjamini–Hochberg multiple test correction. $P$ values of <0.05 were considered statistically significant.

**Reporting summary**. Further information on research design is available in the Nature Research Reporting Summary linked to this article.

## Data availability

Sequence data that support the findings of this study have been deposited in National Center for Biotechnology Information (NCBI) with the primary accession code: PRJNA647523. Source data are available as a Source Data file. The remaining data are available within the article, supplementary information or available from the authors upon request. Source data are provided with this paper.

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

## Acknowledgements

We would like to thank Professor Georgina Hold (Microbiome Research Centre, St George and Sutherland Clinical School, UNSW Sydney); Senior Research Fellow, Dr. Ann Kwan (School of Life and Environmental Sciences, University of Sydney); and Staff Scientist, Dr. Lorna White (School of Life and Environmental Sciences, University of Sydney) for their contribution to this work.

## Author contributions

J.B. contributed to study design, acquisition, analysis, and interpretation of data and manuscript preparation. N.A. contributed to acquisition and interpretation of data and manuscript preparation. X.-T.J. bioinformatic analysis and interpretation of microbiome data. A.R. contributed to acquisition and interpretation of data. L.G. contributed to acquisition and interpretation of data. E.M. contributed to analysis and interpretation of data. R.I. contributed to analysis and interpretation of data. F.C. facilitated subject recruitment, data collection, and manuscript revision. C.S. facilitated subject recruitment and acquisition of data. H.J. facilitated subject recruitment and acquisition of data. V.F. facilitated subject recruitment and acquisition of data. Y.C.K. contributed to acquisition and interpretation of data. M.J. contributed to interpretation of data and manuscript revision. J.O.S. contributed to acquisition, interpretation of data, and manuscript revision. M.W. facilitated subject recruitment, data collection, and manuscript revision. G.M. facilitated subject recruitment, data collection, and manuscript revision. E.E.-O. contributed to study design, acquisition of data, and manuscript revision. A.Z. is the principal investigator who contributed to study design, acquisition, interpretation of data, and manuscript preparation.

## Competing interests

The authors declare no competing interests.
