## [Peer Review File · Nature Communications]

REVIEWER COMMENTS

Reviewer #1 (Remarks to the Author): with expertise in metagenomics

This investigation by Behary and colleagues examines the interplay between the gut microbiome, fecal and circulating SCFAs, fecal metabolites, and peripheral immune responses among a modestly sized cohort of individuals with NAFLD complicated by HCC, NAFLD complicated by cirrhosis, and a healthy control population. This re-demonstrated known associations between gut microbial composition and NAFLD, specifically highlighting enrichment of SCFA producers (and their gene content) in NAFLD-HCC but not NAFLD-cirrhosis, supported by fecal and serum SCFA measurements, and performed ex vivo stimulation studies of peripheral blood mononuclear cells using bacterial extracts derived from the 3-armed study.

The manuscript represents considerable effort on the part of the investigators, and its agreement in microbial ecology with previous studies can be both a strength and a weakness. Its focus on biochemical consequences, particularly SCFA levels in the gut and in circulation, and the differentiation of NAFLD-cirrhosis vs. NAFLD-HCC are all key additions. The immunological results and ex vivo microbial-extract-based phenotyping are strong.

The microbiome analyses and in vivo epidemiology were not always as clear or careful, however, leading to several major comments:

- * The healthy control group was clearly different across many baseline phenotypes from the two NAFLD arms (as expected) in e.g. Table 1, but the authors did not explain how this was accounted for. Patients with metabolic syndrome but without NAFLD, or NAFLD without cirrhosis, would arguably have made more appropriate controls.
- * Likewise the separate microbiome contrasts between NAFLD-cirrhosis and NAFLD-HCC - a potential strength - were not well-investigated. These were often not even noted, as many tests of significance were performed as one way ANOVAs inclusive of the control group (and thus both non-directional and highly significant). Relatedly, as 100% of the NAFLD-HCC patients had concomitant cirrhosis, stark differences between NAFLD-HCC and NAFLD-cirrhosis groups would benefit from further contextualization. Similar contrasts have also been highlighted in other recent NAFLD-related cirrhosis studies (e.g. Oh et al, Cell Metabolism 2020).
- * Specifically, Table 1 demonstrates the recruitment of an older population with compensated end-stage liver disease. Further, the test of difference between experimental groups and healthy controls is less relevant than tests of difference between NAFLD-HCC and NAFLD-cirrhosis. This should be reflected both in the table itself and in the execution and discussion of the associated analyses.
- * The more strongly causal, interventional ex vivo work (with generally clear outcomes) is sometimes conflated with the more ambiguous, correlative, and less quantitatively-justified microbiome epidemiology. I would suggest minimizing the latter overall (particularly as it is largely already reflected in the literature), focusing on the former, and avoiding causal language in vivo (e.g. "we aimed to gain insights into the direct effect of the microbiome on the peripheral immune response in patients with NAFLD-HCC.")
- * This might also help to better-focus the manuscript, which is currently a bit sprawling (10 figures). In general, the Fig. 3-8 results are most core, with Figs. 4-5 and 6-7 sharing information that could be merged / simplified.

Moderate comments:

- * It is not clear in the manuscript how the list of 21 candidate genes involved in SCFA production

was derived. While 5 were found to be differentially abundant, results for the other 16 should be shown in supplementary materials.

* Classifying IL-6 (Figure 8D) as strictly anti-inflammatory is controversial given its role in pro-inflammatory signaling.

* The Discussion could benefit from greater direct comparison with the multitude of large-scale studies linking gut microbial alterations and NAFLD/metabolic syndrome/obesity, including the most aforementioned study by Oh et al.

* While the abstract makes clear that this is a pre-operative cohort, this is not mentioned in the Methods. Greater detail should be offered, particularly with respect to the selection of patients (NAFLD with cirrhosis does not require surgery, NAFLD with HCC resection is most frequently done in patients with small peripheral tumors and good residual synthetic function compared to transplantation, and how the healthy controls were enrolled remains unclear).

* KrakenUniq uses k-mers to ID species-level taxonomic features, so the term OTU should not be used. Several other microbiome analysis best practices are also not followed, e.g. ellipses on ordination Fig. 2B, uninformative phylum level contrasts in Fig. 2C, population averages (without error bars) in both Fig. 2C-D.

Minor comments:

* Several references in the introduction could be clarified (#13) or are off-topic (#14 and #16) and could benefit from a careful re-review.

* Table 2 currently includes rather disparate analyses (that span the entire manuscript) and would benefit from having them separated/restructured.

* Table 2 and Figure 1 are also largely redundant. Perhaps the table could be completely moved to a supplement.

* Greater detail on matching by disease severity should be offered in the Methods (e.g. +/- how many MELD points, +/- BMI, etc.); further, as UNOS allocates liver transplantation on the basis of MELD-Na, this is the preferred clinical scoring system for the assessment of end-stage liver disease.

* A figure detailing study schema may be helpful including enrollment procedure (all eligible consecutive?). This may explain noteworthy Table 1 findings such as the supermajority of males enrolled.

* Table 1 lists the healthy control group as having 81% males. How is this number possible when 24 males out of 30 would be 80% and 25 males would be 83.3%? Similar issues for NAFLD phenotypes.

* AFP in Table 1b should be accompanied by its unit of measure.

Reviewer #2 (Remarks to the Author): with expertise in NAFLD and HCC - clinical

Behary et al conducted this case-control study comparing gut microbiota between 30 patients with cirrhosis, 30 with cirrhosis and HCC, and 30 controls. This study provides novel data linking metagenomic sequencing and metabolomics and how it may impact immune dysfunction that may be relevant to HCC.

I have following key suggestions regarding this paper:

1. The diagnosis of NAFLD is a bit uncertain. Neither elastography nor imaging can provide evidence whether the increased liver stiffness or imaging findings seen are due to NAFLD. It is not likely that authors can go back to these patients so I would perhaps re-classify the patient groups into: Cirrhosis versus cirrhosis-HCC and normal controls not NAFLD related. Unless NAFLD diagnosis can be further bolstered by more detailed description of exclusion of other causes of steatosis.
2. Why are patients with NAFLD related HCC who are cirrhotic undergoing surgical resection? This is not the norm for treatment of such patients as typically liver transplantation is offered to these patients. This requires further details as the study cohort does not represent the entire spectrum of NAFLD related HCC but rather is limited to well-compensated cirrhotics who have early HCC. Is the study conducted in a non-transplant center?
3. Is there an association between Cirrhosis with HCC and degree of hepatic decompensation- e.g. with presence of ascites?
4. Is it possible that the changes that are seen are due to liver resection leading to change/reduction in total bile acid pool and changes in microbiome/metabolome? This should be discussed in limitations/perspectives.
5. To address the issue of resection would these changes we observed in those who undergo resection for other causes such as a large adenoma?
6. Are there any data that these changes are specific to NAFLD cirrhosis and HCC and not seen with cirrhosis versus cirrhosis with HCC of any etiology?
7. Fecal microbiota profile in cirrhosis and HCC appears to have some commonality between recent studies in similar populations such as Qin et al. Nature 2014, Loomba et al. Cell Metabolism 2017, Caussy et al. Nature communications 2019, and Oh et al. Cell metabolism 2020. These should be discussed e.g. Veillonella, Ruminococcus and Clostridia appears to be a significant discriminator.
8. Findings related to SCFA and metabolomics are particularly interesting and add to the novelty of the work.
9. I found the work shown in figure 9 particularly novel linking specific species with possible immune dysregulation that opens up new areas for hypothesis testing and perhaps interventions in future.
10. How many patients had diabetes? Did that impact the results? How many patients were on metformin for diabetes, lactulose for HE and did those impact the gut microbiome?
11. When were fecal samples post resection collected? Was there a difference based upon the timing of stool specimen collection and findings given acute changes in bile acid metabolism.
12. Are there any data on the total serum bile acid profile in these cohorts?
13. A recent study in Hepatology in press Ling et al. Showed that FGF-19 led to an increase in Veillonella in patients with NASH in a clinical trial. It may have some relevance with your findings. As Veillonella may be sensitive to inhibition of bile acid synthesis post resection and may be a marker of response. Do you have samples prior to resection and post resection and what type of changes you observe?

Overall, an excellent effort by a respected multi-disciplinary team, I am very enthusiastic of the direction of research and the work done in this manuscript, and hopefully above points would be helpful in further improving the quality of the content. Keep it up!

With best wishes,
Rohit Loomba

Reviewer #3 (Remarks to the Author): with expertise in immunometabolism and cancer

Behary et al. employ metagenomic and metabolomic methodology to explore the impact of microbiome and microbiome-derived metabolites on immune responses in NAFLD-HCC and NAFLD-cirrhosis patients compared to healthy controls. The authors confirm dysbiosis in both disease

cohorts and identify elevated levels of short-chain fatty acids such as butyrate, which they show to enhance Treg proliferation and reduce CD8 T cell numbers. The study uses generally well-characterized cohorts to address pertinent questions about the interactions of the microbiome with the host immune system in a disease context of high relevance. Yet, shortcomings cast doubt on important conclusions of this study.

Major concerns:

+ In several instances, novelty of the results is unclear. For example, Ren Z et al. Gut 2019 (not cited by Behary et al.) described the microbiome of HCC patients and compared it to cirrhotic and healthy controls. In contrast to the present study, that study found butyrate-producing genera decreased in HCC versus controls. The impact of SCFAs on Treg differentiation was previously reported and so was the role of SCFAs in HCC (e.g. Singh V et al. Cell 2018). Also, the authors mention that high-fiber diet increases concentrations of SCFA in serum and feces but fail to cite important studies (e.g. Trompette et al. Immunity, 2018).

+ The authors do not disclose their list of 21 microbial candidates responsible for SCFA synthesis (Fig. 3, lines 434 - 440). This could be shown e.g. within Fig. 3A together with a more comprehensive overview of the respective metabolic pathways, indicating rate-limiting steps respectively genes whose expression determines the flux through this pathway. Likewise, the authors do not disclose their list of 65 metabolites "important in bacterial metabolic processes", which makes it impossible to evaluate their claims that they are representative for bacterial metabolic processes.

+ Fig. 5/6/7: The authors conclude that butyrate is the driver behind the observed Treg expansion and the diminished populations of cytotoxic CD8+ T cell in NAFLD-HCC patients. However, due to the lack of disclosure of the full panel of measured metabolites nor the complete results from this metabolomics panel, it remains unclear whether other metabolites may show similar correlations. In each current form the study appears to be biased towards selected metabolites.

+ In vitro dose titration experiments with butyrate and/or bacterial extracts would strengthen the study's conclusions regarding the described immunosuppressive effects.

+ Cellular readouts rely mostly on proliferation. Additional data on cytokine production and activation markers respectively functional assays for monocytes would help to better characterize the immunosuppressive effects.

+ The authors describe a positive correlation between numbers of Treg cells/IL-10+ Treg cells and butyrate concentrations in feces and a negative correlation between numbers of CD8+ T cell and butyrate concentrations in feces (Fig. 9). How do the authors reconcile these observations to those made in other publications that show a stimulating effect of butyrate on CD8+ T cell expansion and effector function (e.g. Trompette et al. Immunity, 2018)?

Minor concerns:

+ Is it known whether the cohorts differ in their diets and food intake? Considering the literature on SCFA, this would be useful additional information.

+ The authors prepared bacterial extracts from stool samples of their study cohorts. What were the criteria to select 10 patients from each cohort for these experiments?

+ Fig 9: Color changes are difficult to recognize. It is also unclear why a heatmap is shown for the presented small number of data points.

REVIEWER COMMENTS

Reviewer #1 (Remarks to the Author): with expertise in metagenomics

This investigation by Behary and colleagues examines the interplay between the gut microbiome, fecal and circulating SCFAs, fecal metabolites, and peripheral immune responses among a modestly sized cohort of individuals with NAFLD complicated by HCC, NAFLD complicated by cirrhosis, and a healthy control population. This re-demonstrated known associations between gut microbial composition and NAFLD, specifically highlighting enrichment of SCFA producers (and their gene content) in NAFLD-HCC but not NAFLD-cirrhosis, supported by fecal and serum SCFA measurements, and performed *ex vivo* stimulation studies of peripheral blood mononuclear cells using bacterial extracts derived from the 3-armed study.

The manuscript represents considerable effort on the part of the investigators, and its agreement in microbial ecology with previous studies can be both a strength and a weakness. Its focus on biochemical consequences, particularly SCFA levels in the gut and in circulation, and the differentiation of NAFLD-cirrhosis vs. NAFLD-HCC are all key additions. The immunological results and *ex vivo* microbial-extract-based phenotyping are strong.

The microbiome analyses and in vivo epidemiology were not always as clear or careful, however, leading to several **major comments**:

** The healthy control group was clearly different across many baseline phenotypes from the two NAFLD arms (as expected) in e.g. Table 1, but the authors did not explain how this was accounted for. Patients with metabolic syndrome but without NAFLD, or NAFLD without cirrhosis, would arguably have made more appropriate controls.*

It is expected for the NAFLD-cirrhosis and NAFLD-HCC cohorts (NAFLD arms) to have a higher prevalence of metabolic disease (type II diabetes, hypertension and dyslipidaemia) than healthy controls (Table 1). We appreciate that a control group with the same rates of metabolic disease would also have served as an appropriate control cohort. However, in our study, we matched the groups for age, sex and BMI; all established to impact microbiome composition. In clinical practice, attempting to recruit a modestly sized cohort of healthy control, matched for the same age group as our NAFLD cirrhosis cohort, with diabetes, but without a degree of NAFLD would be challenging, as the presence of metabolic disease, in particular diabetes in this population substantially increases the prevalence of NAFLD, NAFLD fibrosis and cirrhosis (Angulo *et al*, Hepatology, 1999 and Loomba *et al*, Hepatology 2012). This point has been discussed and addressed under the limitation section of the discussion.

** Likewise, the separate microbiome contrasts between NAFLD-cirrhosis and NAFLD-HCC - a potential strength - were not well-investigated. These were often not even noted, as many tests of significance were performed as one-way ANOVAs inclusive of the control group (and thus both non-directional and highly significant). Relatedly, as 100% of the NAFLD-HCC patients had concomitant cirrhosis, stark differences between NAFLD-HCC and NAFLD-cirrhosis groups would benefit from further contextualization. Similar contrasts have also been highlighted in other recent NAFLD-related cirrhosis studies (e.g. Oh *et al*, Cell Metabolism 2020).*

In our first submission, we attempted to illustrate two group comparisons (between NAFLD cirrhosis and NAFLD-HCC) in Figure 2c-d by denoting symbols for these respective comparisons, as is described in the figure legend: ‘# $P < 0.05$ NAFLD-HCC compared to healthy control, † $P < 0.05$ NAFLD-HCC compared to NAFLD-cirrhosis and ‡ $P < 0.05$ NAFLD-cirrhosis compared to healthy control’. We acknowledge now that this is not clear to readers, and thus, have highlighted important differences between groups in the results and discussion of the manuscript, and have provided **detailed data** with 3 group and 2 group comparisons in Supplementary Table 2. Further, we have provided

context to our findings, highlighting the similarities in our microbiota signature in NAFLD-cirrhosis to that recently published by Oh *et al*, Cell Metabolism 2020.

**** Specifically, Table 1 demonstrates the recruitment of an older population with compensated end-stage liver disease. Further, the test of difference between experimental groups and healthy controls is less relevant than tests of difference between NAFLD-HCC and NAFLD-cirrhosis. This should be reflected both in the table itself and in the execution and discussion of the associated analyses.***

We appreciate the reviewer's comment. Importantly, we did **not** identify a significant difference in any of the variables (shown in Table 1) between the NAFLD-cirrhosis and NAFLD-HCC groups. We have included this in the Table 1 legend and the results section of the manuscript for clarity. In addition, differences in variables between liver disease groups and healthy controls are denoted in Table 1 as indicated by the symbol '*'. We have revised the results section pertaining to this for clarity.

**** The more strongly causal, interventional ex vivo work (with generally clear outcomes) is sometimes conflated with the more ambiguous, correlative, and less quantitatively justified microbiome epidemiology. I would suggest minimizing the latter overall (particularly as it is largely already reflected in the literature), focusing on the former, and avoiding causal language in vivo (e.g. "we aimed to gain insights into the direct effect of the microbiome on the peripheral immune response in patients with NAFLD-HCC.")***

We appreciate the reviewer comment that our findings of our *ex vivo* model are novel, more so than the microbiome findings that have been reported previously using 16S rRNA analyses. The novelty of the microbiome data in our study is the fact that it is the first metagenomics comparison of NAFLD-HCC versus NAFLD-cirrhosis. We agree that the statement "we aimed to gain insights into the direct effect of the microbiome on the peripheral immune response in patients with NAFLD-HCC" insinuates causality and have deleted the statement

**** This might also help to better focus the manuscript, which is currently a bit sprawling (10 figures). In general, the Fig. 3-8 results are most core, with Figs. 4-5 and 6-7 sharing information that could be merged / simplified.***

We have merged figures 4-5 and 6-7 which share information to better focus the manuscript.

Moderate comments:

**** It is not clear in the manuscript how the list of 21 candidate genes involved in SCFA production was derived. While 5 were found to be differentially abundant, results for the other 16 should be shown in supplementary materials.***

The 21 candidate genes involved in SCFAs production were derived from previously published literature, in particularly a study by Zhao *et al*, Nature Partner Journals – Biofilms and Microbes, 2019 which reported genes well known to be involved in SCFA synthesis, in addition to those that have been reported more recently. Relevant references have been included in the methods to highlight these papers. Also, as requested, a table of results demonstrating details regarding differential gene abundance of all 21 candidate genes, along with all 3 group and all 2 group comparisons has been included for clarity (Supplementary Table 3).

**** Classifying IL-6 (Figure 8D) as strictly anti-inflammatory is controversial given its role in pro-inflammatory signalling.***

We have amended this section of the results to reflect this statement.

** The Discussion could benefit from greater direct comparison with the multitude of large-scale studies linking gut microbial alterations and NAFLD/metabolic syndrome/obesity, including the most aforementioned study by Oh et al.*

We have amended and contextualised the discussion in relation to published papers in the area.

** While the abstract makes clear that this is a pre-operative cohort, this is not mentioned in the Methods. Greater detail should be offered, particularly with respect to the selection of patients (NAFLD with cirrhosis does not require surgery, NAFLD with HCC resection is most frequently done in patients with small peripheral tumors and good residual synthetic function compared to transplantation, and how the healthy controls were enrolled remains unclear*

We have included all details pertaining to this in the methods section of the manuscript and have provided information regarding the recruitment of healthy controls in Supplementary materials.

** KrakenUniq uses k-mers to ID species-level taxonomic features, so the term OTU should not be used. Several other microbiome analysis best practices are also not followed, e.g. ellipses on ordination Fig. 2B, uninformative phylum level contrasts in Fig. 2C, population averages (without error bars) in both Fig. 2C-D.*

We appreciate that the term OTU should not be used in reference in KrakenUniq output and have adjusted Figure 2a and the results to ‘observed number of species’ to more accurately reflect this measure of alpha-diversity. With regard to the constrain analysis of principle (CAP) illustrated in Figure 2b, the ellipses that are drawn as done so using `stat_ellipse` (`geom = "polygon", type="norm", alpha=0, level=0.95`) in `ggplot2` to reflect separation of groups. We agree that clarity is required around this point and have included this detail in the methods section of the manuscript and Figure 2 legend. With regard to Figure 2c and d, we chose to represent variation in taxonomy as stacked bar plots, we originally attempted to illustrate two group comparisons calculated by Kruskal-Wallis with Dunn’s post hoc test in Figure 2c-d by denoting symbols for these respective comparisons, as is described in the figure legend: ‘# $P < 0.05$ NAFLD-HCC compared to healthy control, † $P < 0.05$ NAFLD-HCC compared to NAFLD-cirrhosis and ‡ $P < 0.05$ NAFLD-cirrhosis compared to healthy control’. For clarity, we have highlighted important differences between groups in the results and discussion of the manuscript and **have provided detailed taxonomic data with 3 group and all 2 group comparisons in Supplementary Table 2**. Additionally, we performed LEfSe analysis to identify differentially abundant taxa and have provided this information in Supplementary Figure 4 and Supplementary Figure 5.

Minor comments:

** Several references in the introduction could be clarified (#13) or are off-topic (#14 and #16) and could benefit from a careful re-review.*

We have amended the reference. With respect to references 14 and 16, we would respectfully like to include these, as although they are not directly related to liver disease, these studies do highlight important interactions between the microbiome, its metabolites and T cell immunity which we feel are relevant to our work.

** Table 2 currently includes rather disparate analyses (that span the entire manuscript) and would benefit from having them separated/restructured.*

We have removed Table 2, and have replaced this with separate tables, Supplementary Tables 1 – 5, organised by analysis type such as microbiota data, metabolomic data, immune data etc. In addition, in these tables, we have provided detail, including all 3 group and 2 group comparisons for clarity.

** Table 2 and Figure 1 are also largely redundant. Perhaps the table could be completely moved to a supplement.*

We have removed original Table 2 and included relevant tables of data in Supplementary material. We feel that it is important to demonstrate the baseline immune phenotype in our patient cohort as this is relevant to the findings of our *ex vivo* model and would prefer for Figure 1 to remain in the manuscript if acceptable

** Greater detail on matching by disease severity should be offered in the Methods (e.g. +/- how many MELD points, +/- BMI, etc.); further, as UNOS allocates liver transplantation on the basis of MELD-Na, this is the preferred clinical scoring system for the assessment of end-stage liver disease.*

We have included details on matching including age, gender, BMI, and extent of liver disease parameters (Child Pugh and MELD-Na score) in the methods and results section of the manuscript. We accept that the United Network of Organ Sharing uses MELD-Na score to stage for end-stage liver disease, and thus have recalculated the MELD to MELD-Na scores as suggested. This result is presented in the manuscript and in Table 1.

** A figure detailing study schema may be helpful including enrolment procedure (all eligible consecutive?). This may explain noteworthy Table 1 findings such as the supermajority of males enrolled.*

We thank the reviewer for this comment. To avoid the addition of further figures, as was suggested in previous comment, we have detailed in the recruitment process in the methods section.

** Table 1 lists the healthy control group as having 81% males. How is this number possible when 24 males out of 30 would be 80% and 25 males would be 83.3%? Similar issues for NAFLD phenotypes.*

We have reviewed the data and made corrections to Table 1.

** AFP in Table 1b should be accompanied by its unit of measure.*

We have added ng/mL as the unit of measure for AFP. To comply with Nature Communications formatting with respect to Tables, we have re-named the original Table 1b to Table 2. Thus, we have added this detail to Table 2.

Reviewer #2 (Remarks to the Author): with expertise in NAFLD and HCC - clinical

Behary *et al* conducted this case-control study comparing gut microbiota between 30 patients with cirrhosis, 30 with cirrhosis and HCC, and 30 controls. This study provides novel data linking metagenomic sequencing and metabolomics and how it may impact immune dysfunction that may be relevant to HCC. I have following key suggestions regarding this paper:

1. The diagnosis of NAFLD is a bit uncertain. Neither elastography nor imaging can provide evidence whether the increased liver stiffness or imaging findings seen are due to NAFLD. It is not likely that authors can go back to these patients so I would perhaps re-classify the patient groups into: Cirrhosis versus cirrhosis-HCC and normal controls not NAFLD related. Unless NAFLD diagnosis can be further bolstered by more detailed description of exclusion of other causes of steatosis.

The diagnosis of NAFLD-cirrhosis was made based on clinical, biochemical, transient elastography, and radiological assessment in patients who had metabolic disease, and in the absence of other causes of hepatic steatosis as per American Association for the Study of Liver Diseases (AASLD) practice guidelines, Chalasani *et al* Hepatology 2012. Such exclusion criteria included current or previous excess alcohol intake (≥ 30 g daily for men and ≥ 20 g daily for women) and other causes of liver disease including viral, alcoholic, autoimmune, cholestatic liver diseases and inherited liver diseases. We have included this detail in the methods section of the manuscript.

2. *Why are patients with NAFLD related HCC who are cirrhotic undergoing surgical resection? This is not the norm for treatment of such patients as typically liver transplantation is offered to these patients. This requires further details as the study cohort does not represent the entire spectrum of NAFLD related HCC but rather is limited to well-compensated cirrhotics who have early HCC. Is the study conducted in a non-transplant center?*

Patients recruited to this study had HCC, staged with BCLC as stage 0 or A, with preserved liver function and absence of portal hypertension (Table 1 and Table 2). Based on this, liver resection is recommended by international consensus groups, including the American Association for the Study of Liver Diseases (AASLD), Marrero *et al*, Hepatology, 2018. We have provided clarity around this point in the methods and results sections of the manuscript.

3. *Is there an association between Cirrhosis with HCC and degree of hepatic decompensation- e.g. with presence of ascites?*

Patients included in the study had Child Pugh A liver disease and **no** evidence of portal hypertension (Table 1), and thus did not have any degree of hepatic decompensation such as variceal bleeding, hepatic encephalopathy or ascites. We have provided clarity around this point in the methods and results sections of the manuscript.

4. *Is it possible that the changes that are seen are due to liver resection leading to change/reduction in total bile acid pool and changes in microbiome/metabolome? This should be discussed in limitations/perspectives.*

In this study, patients with NAFLD-HCC had their study samples (including blood and microbiome samples) collected in the week **prior** to surgery. We have provided clarity to this in the methods section. We believe that it would be relevant to examine how surgery impacts on microbiome/metabolome alterations in patients with HCC, which is among the aspects of our current HCC related work.

5. *To address the issue of resection would these changes we observed in those who undergo resection for other causes such as a large adenoma?*

Great question!- Following from response to comment 4, in this study, patients with NAFLD-HCC had their study samples (including blood and microbiome samples) collected in the week **prior** to surgery. We agree that in addition to surgery itself, the impact of varying indications of surgery on alterations of the microbiome/metabolome would be exciting to study and would require a larger cohort of patients (not exclusive to HCC) and post-resection blood and faecal samples.

6. *Are there any data that these changes are specific to NAFLD cirrhosis and HCC and not seen with cirrhosis versus cirrhosis with HCC of any etiology?*

Our study is limited by the lack of a non-NAFLD cirrhosis/HCC cohort that would allow us to confirm that microbiome/metabolomic changes are not seen with liver disease of other aetiology. Our findings were different to those by Ren *et al*, whom compared to our study, used relatively younger cohort of patients, with lower BMI and with liver disease due to chronic hepatitis B. The differences in the microbiome signature could plausibly be due to the underlying liver disease, however, this could not be confirmed without head to head comparison. We have included the paper by Ren *et al* in the discussion

and discussed their findings in relation to ours, and the need for large cohort studies to examine the role of the underlying liver disease on the HCC related microbiome.

7. Fecal microbiota profile is in cirrhosis and HCC appears to have some commonality between recent studies in similar populations such as Qin et al. Nature 2014, Loomba et al. Cell Metabolism 2017, Caussy et al. Nature communications 2019, and Oh et al. Cell metabolism 2020. These should be discussed e.g. Veillonella, Ruminococcus and Clostridia appears to be a significant discriminator.

Indeed, we have included this important point in our discussion. Our study has identified a microbiota signature in NAFLD-cirrhosis very similar to that previously published. Despite the use of different control groups, human metagenomic data from our cohort and others support an emerging core microbiome signature that characterises NAFLD-cirrhosis, with enrichment of *Ruminococcus gnavus*, *Clostridium bolteae*, *Streptococcus parasanguinis* and *Klebsiella pneumoniae*, and reduced abundance of beneficial species including *Faecalibacterium prausnitzii*, *Alistipes putredinis*, and *Eubacterium eligens*. This is important, as it demonstrates the potential utility of the microbiome as a diagnostic tool in NAFLD-cirrhosis.

Similarly, in HCC, we detected enrichment in *Bacteroides* and *Ruminococcaceae* taxa, previously identified by Ponziani *et al*, Hepatology 2019 to correlate with several systemic inflammatory and immune markers. We have provided contextualisation to our findings in the discussion of our paper and are excited to see liver disease/HCC microbiota signatures are consistent despite geographical differences. This is another strength to our study.

8. Findings related to SCFA and metabolomics are particularly interesting and add to the novelty of the work.

We appreciate this comment provided by the reviewer.

9. I found the work shown in figure 9 particularly novel linking specific species with possible immune dysregulation that opens up new areas for hypothesis testing and perhaps interventions in future.

Again, we greatly appreciate this comment provided by the reviewer and we are working on experiments pertaining to this comment.

10. How many patients had diabetes? Did that impact the results? How many patients were on metformin for diabetes, lactulose for HE and did those impact the gut microbiome?

We thank the reviewer for this comment. Data on the prevalence of type II diabetes in the three study groups is shown in Table 1. There was no difference in prevalence of type II diabetes between the NAFLD-cirrhosis (50%) and NAFLD-HCC groups (50%) (Table 1). Additionally, there was no difference in rates of metformin use between the groups; 39.3% of patients with NAFLD-cirrhosis and 37.5% of patients with NAFLD-HCC were on metformin. This additional data has been added to Table 1 as per reviewer's suggestion. Importantly, despite these groups being not statistically different with respect to type II diabetes and metformin use, significant changes were seen between these groups with respect to the microbiome, metabolome and elicited responses seen in our *ex vivo* model.

Our patients with liver disease (NAFLD-cirrhosis and NAFLD-HCC) has Child Pugh A stage liver disease with the absence of portal hypertension, and thus did not have any prior episode of documented hepatic decompensation such as hepatic encephalopathy (HE), variceal bleeding or ascites. Thus, none of these patients were on lactulose therapy.

11. When were fecal samples post resection collected? Was there a difference based upon the timing of stool specimen collection and findings given acute changes in bile acid metabolism.

All faecal samples in this study were collected in the week **prior** to resection. We have provided clarity to this point in the methods section of the manuscript. We do however appreciate a comparative microbiome analysis and bile acid profiling of patients pre and post liver resection would be an interesting study. We have highlighted this aspect in the limitation section of the discussion.

12. Are there any data on the total serum bile acid profile in these cohorts?

We thank the reviewer for this question and understand the importance of total serum bile acid analysis within the context of liver resection, as was discussed in point 11. We do not have this data available, but do agree that bile acid analysis, along with microbiome analysis pre- and post- liver resection in the NAFLD-HCC cohort would be an interesting study to perform. This will be one of the aspects of our future work.

13. A recent study in Hepatology in press Ling et al. Showed that FGF-19 led to an increase in Veillonella in patients with NASH in a clinical trial. It may have some relevance with your findings. As Veillonella may be sensitive to inhibition of bile acid synthesis post resection and may be a marker of response. Do you have samples prior to resection and post resection and what type of changes you observe?

We have read with interest the outstanding work by Ling *et al.* The reviewer raises an important concept regarding the prevalence of *Veillonella* in patients with NAFLD-HCC. Our patients were consecutively recruited into our study, and an adequate follow – up period had not yet occurred to assess response to treatment. In our patient cohort with NASH, and with low alpha-foetoprotein. Over the next 5 years, we will be closely documenting rates of relapse of our patients who have undergone ‘curative’ treatment of their HCC, and we may identify relevant and important relationships between *Veillonella* and treatment / relapse rates in our study cohort. We appreciate this important consideration and will include this in our planning of forthcoming studies.

Reviewer #3 (Remarks to the Author): with expertise in immunometabolism and cancer

Behary *et al.* employ metagenomic and metabolomic methodology to explore the impact of microbiome and microbiome-derived metabolites on immune responses in NAFLD-HCC and NAFLD-cirrhosis patients compared to healthy controls. The authors confirm dysbiosis in both disease cohorts and identify elevated levels of short-chain fatty acids such as butyrate, which they show to enhance Treg proliferation and reduce CD8 T cell numbers. The study uses generally well-characterized cohorts to address pertinent questions about the interactions of the microbiome with the host immune system in a disease context of high relevance. Yet, shortcomings cast doubt on important conclusions of this study.

Major concerns:

+ In several instances, novelty of the results is unclear. For example, Ren Z et al. Gut 2019 (not cited by Behary et al.) described the microbiome of HCC patients and compared it to cirrhotic and healthy controls. In contrast to the present study, that study found butyrate-producing genera decreased in HCC versus controls. The impact of SCFAs on Treg differentiation was previously reported and so was the role of SCFAs in HCC (e.g. Singh V et al. Cell 2018). Also, the authors mention that high-fiber diet increases concentrations of SCFA in serum and feces but fail to cite important studies (e.g. Trompette et al. Immunity, 2018).

A) We have discussed the disparate microbiota findings in our study compared to that published by Ren *et al.*, Gut 2019. The contrasting findings are plausibly due to significant differences in

the study cohorts and control groups. In the study by Ren *et al*, chronic hepatitis B infection was the aetiology of the underlying liver disease in the studied cohort who additionally had a much lower BMI (22.8 ± 2.04 kg/m²), compared to our study cohort with a high BMI (31.5 ± 7.6 kg/m²) and many metabolic risk factors. Additionally, Ren *et al*, used 16S rRNA sequencing to ‘predict’ the function of genera enriched in their HCC cohort, whilst we used shotgun metagenomic sequencing to compare abundance of genes related to SCFA synthesis between groups followed by quantification of SCFAs in faecal samples from these patients. The paper by Ren *et al*, is a landmark paper in the field. Thus, we have included this citation and discussed our findings in relation to theirs.

- B) We appreciate that the impact of SCFAs on Treg differentiation and the role of SCFAs in HCC have been previously reported in animal models. Our findings bring these important concepts together in a human cohort, adding to the body of evidence that these mechanisms may have important clinical implications in patients with NAFLD-HCC. We have included these important papers as citations, and again have discussed our findings in the context of these established literature in the discussion section.

+ The authors do not disclose their list of 21 microbial candidates responsible for SCFA synthesis (Fig. 3, lines 434 - 440). This could be shown e.g. within Fig. 3A together with a more comprehensive overview of the respective metabolic pathways, indicating rate-limiting steps respectively genes whose expression determines the flux through this pathway. Likewise, the authors do not disclose their list of 65 metabolites “important in bacterial metabolic processes”, which makes it impossible to evaluate their claims that they are representative for bacterial metabolic processes.

The 21 candidate genes involved in SCFA production was derived from previously published literature, in particularly a study by Zhao *et al*, Nature Partner Journals – Biofilms and Microbes, 2019 which reported genes well known to be involved in SCFA synthesis, in addition to those that have been reported more recently. **For clarity**, a table of results demonstrating the differential gene abundance of all 21 candidate genes, along with all 3 group and all 2 group comparisons has been included for clarity (Supplementary Table 3). Likewise, a detailed table of results of all 65 measured metabolites in addition to SCFAs, along with 3 group and all 2 group comparisons have been included in Supplementary Table 4.

+ Fig. 5/6/7: The authors conclude that butyrate is the driver behind the observed Treg expansion and the diminished populations of cytotoxic CD8+ T cell in NAFLD-HCC patients. However, due to the lack of disclosure of the full panel of measured metabolites nor the complete results from this metabolomics panel, it remains unclear whether other metabolites may show similar correlations. In each current form the study appears to be biased towards selected metabolites.

We thank the reviewer and appreciate that presenting only those metabolites where alterations in NAFLD-HCC group were seen, appear to demonstrate bias towards selected metabolites. We have made several changes to address this. In Supplementary Figure 6 and for ease of read, we have only included results of metabolites where a significant change was seen between groups. For detailed data on all metabolites, we have included a Supplementary Table 4, which presents a complete list of all measured metabolites, and all data pertaining to this, including all 3 group and 2 group comparisons. Finally, in Supplementary Figure 8, we have illustrated correlations between **all** measured metabolites against T-cell responses as measured in our *ex vivo* model.

+ In vitro dose titration experiments with butyrate and/or bacterial extracts would strengthen the study’s conclusions regarding the described immunosuppressive effects.

We thank the reviewer for this insightful comment. As the dose of bacterial extracts (BE) was not included in the manuscript that originally described the *ex vivo* model (Cekanaviciute E *et al* PNAS 2017), our initial experimental approach was to assess titrating doses of BE on PBMC viability so as not to compromise T cell responses. To this end, we commenced our experiment with an arbitrarily

selected BE dose of 50mg/mL but found this to significantly impact PBMC viability. Thus, with sequential down titration of BE dose, we found 10mg/mL to be the maximal dose of BE whereby PBMC viability was maintained at > 90%. For this reason, titration experiments for BE on T cell responses were not feasible. Data pertaining to this experiment is shown in Supplementary Fig 1.

With respect to SCFAs, particularly butyrate, previous *ex vivo* experiments with PBMCs, have already shown that butyrate alone and in a dose dependant manner can influence T cell responses (Asarat, M, *et al*, Immunol Invest, 2016 & Zhang, M *et al*, BMC Gastroenterol 2016). However, the **isolated** *ex vivo* effect of SCFAs (or butyrate) on PBMCs was not the focus of, or the message from, the current study. Rather through the combined *ex vivo* model and the correlative studies, we illustrate the importance of the **composition** of the dysbiotic microbiome and its collective metabolites in NAFLD-HCC as the main driver of the observed *ex vivo* T cell responses. Arguably, this is a more realistic, real-time and powerful approach rather than using single SCFAs. We have addressed this in the discussion. Notably, our key conclusion in the abstract was that it was the collective dysbiosis phenotype (microbiome and related metabolites) that elicited the detected immunosuppressive response.

+ Cellular readouts rely mostly on proliferation. Additional data on cytokine production and activation markers respectively functional assays for monocytes would help to better characterize the immunosuppressive effects.

In our study we measured 48 cytokines and chemokines in our *ex vivo* model. Based on this recommendation from the reviewer, we have included detailed data pertaining to these 48 cytokines, including their measured concentration, as well as 3 group and 2 group comparisons in Supplementary Table 5.

+ The authors describe a positive correlation between numbers of Treg cells/IL-10+ Treg cells and butyrate concentrations in feces and a negative correlation between numbers of CD8+ T cell and butyrate concentrations in feces (Fig. 9). How do the authors reconcile these observations to those made in other publications that show a stimulating effect of butyrate on CD8+ T cell expansion and effector function (e.g. Trompette *et al*. Immunity, 2018)?

We appreciate that the study by Trompette *et al*, Immunity, 2018 demonstrated the stimulating effect of butyrate on CD8+ T cells, which is in contrast to correlations that we identified in our study between CD8+ T cells and butyrate concentrations. This is certainly an important finding.

The effect of butyrate on CD8+ T cell responses are certainly complex and do not appear to be unidirectional. For example, Uribe-Herranz, M. *et al*. J Clin Invest 2020 demonstrated that vancomycin mediated elimination of bacteria-producing SCFAs was accompanied by increased cytotoxic T cell tumour infiltration. Furthermore, inhibition of CD8+ T cell responses in our study may have occurred secondary to butyrate mediated Treg expansion and enhanced function. Thus, butyrate is reported to mediate expansion of Treg populations in PBMCs through enhancement of FoxP3 expression (Asarat *et al*, Immunological Investigations, 2016 and Zhang *et al*, BMC Gastroenterology, 2016) and Tregs have been shown, through competition for IL-2 to reduce CD8+ T cell expansion (Hofer *et al*, Frontiers in Immunology, 2012). Additionally, microbiota mediated SCFAs have been shown to skew CD8+ T cells towards a memory phenotype in cell culture models (Bachem *et al*, Immunity 2019).

We have discussed the complex interaction between butyrate and CD8+ T cells in the discussion section of the manuscript.

Minor concerns:

+ Is it known whether the cohorts differ in their diets and food intake? Considering the literature on SCFA, this would be useful additional information.

All patients recruited to the study were on a mixed diet, as has been reported in previous studies such as Ren *et al*, Gut 2019. This detail has been added to Table 1a. Similar to other studies in the field, a

comprehensive dietary analysis was not performed on patients recruited to this study. This will be included in future work.

+ *The authors prepared bacterial extracts from stool samples of their study cohorts. What were the criteria to select 10 patients from each cohort for these experiments?*

These BE were prepared from the most dysbiotic faecal samples, based on alpha-diversity in NAFLD-HCC (3838 ± 881.0 , mean number of observed species) and NAFLD-cirrhosis (3984.1 ± 288.5) and least dysbiotic faecal samples in healthy control group (5296.3 ± 121.9). We have included this statement in the results section of the manuscript.

+ *Fig 9: Color changes are difficult to recognize. It is also unclear why a heatmap is shown for the presented small number of data points.*

We have thus adjusted the heat map colour scale so that colour changes are easier to appreciate. We have included all significant and non-significant correlations and have denoted significant correlations with a “+”, as was done in Oh *et al*, Cell Metabolism 2020. Also, we have added Supplementary Figure 8, to illustrate correlations between **all** measured metabolites against T-cell responses as measured in our *ex vivo* model.

REVIEWERS' COMMENTS

Reviewer #1 (Remarks to the Author):

In the revision by Behary and colleagues, the authors have significantly improved the manuscript's text in response to our comments, streamlined their findings, and have placed their results in the context of the other considerable works in the field. A few minor comments remain:

- * In response to our first previous point (phenotypic differences with respect to the NAFLD arms), we agree this is expected, and difficult to address clinically (as we pointed out). The concern was instead that these differences aren't accounted for in the analysis. Rather than just pointing out that recruiting perfect controls would be impossible, it would be more appropriate to either 1) truly incorporate these limitations into the analysis (i.e. via additional covariates), or at least 2) acknowledge the degree to which the current analysis cannot distinguish causality or differences among these confounding effects.
- * Likewise the concern in our second previous point (NAFLD-cirrhosis vs. NAFLD-HCC differences) was not that the comparison was not performed, but that the results - despite being very interesting - were not analyzed. This appears to still be the case, with the quantitative results relegated to the supplement with no corresponding main text. Likewise we were not able to find any quantitative or systematic comparison with the previous biomarker presented, just a qualitative mention of several shared microbes in the Discussion.
- * The figure improvements are a great help both for visualization and for quantitative clarity. Likewise the minor text, Methods, and reference updates are all helpful.
- * In the section detailing experiments stimulating PBMCs with bacterial extracts selected from "the most dysbiotic samples," was this intended to indicate selection from the least diverse (as dysbiotic and least diverse are not synonymous)?
- * In the same section, the mean (SD) for number of observed species is in the mid-3,000s vs. 5,000s for controls. This appears to be a typographical or transcribing error, but if not warrants further explanation, as even the most deeply sequenced human microbiome samples typically have species counts on the order of ~low hundreds. This elevated count is also observed in Fig. 2A. These counts might be describing strain variants or some other type of sub-species clade instead?
- * In Table 1, if no further diet information can be provided beyond simply "mixed," it could be omitted and replaced with text simply stating that these were all free-living individuals consuming ad-libitum diets.
- * In Supplementary Fig. 4, the inclusion of essentially all taxonomic ranks (i.e. species, genus, family, etc.) in the LDA leads to a laundry list of significant associations, many of which are duplicative (Fig. 4B and clearest in clades with few branches e.g. Fusobacteriaceae). It may be useful to visualize only the species level given the metagenomic approach and lack of biologically actionable genus-level (or higher) associations.

Reviewer #3 (Remarks to the Author):

The authors adequately addressed all my concerns.

Reviewer #4 (Remarks to the Author):

I have read with great interest the revised manuscript of the study performed by Behary et al.

The revised version have been improved and the authors have satisfactory addressed the majority of the comment from the reviewer #2.

There is still an issue regarding the classification of the study population regarding the control group as pointed by the reviewer #2 point 1. Indeed, based upon the supplemental methods, the participants of the control group were carefully screened for other causes of liver disease than NAFLD. "In addition, among the control group 13,3% had type 2 diabetes mellitus, 16,6% had hypertension and 10% had dyslipidemia in Table 1. Hence, these participants are not "healthy" but non-NAFLD controls. This reviewer agree with the reviewer 2 and suggest to rename the "healthy group" as "non-NAFLD group" along the text, figure and Table for better accuracy as suggested initially by the reviewer 2.

Dr Cyrielle CAUSSY

Please see below point by point response to the reviewers' comments/queries. We thank the reviewers for their contribution to improving the manuscript.

REVIEWERS' COMMENTS

Reviewer #1 (Remarks to the Author):

In the revision by Behary and colleagues, the authors have significantly improved the manuscript's text in response to our comments, streamlined their findings, and have placed their results in the context of the other considerable works in the field. A few minor comments remain:

** In response to our first previous point (phenotypic differences with respect to the NAFLD arms), we agree this is expected, and difficult to address clinically (as we pointed out). The concern was instead that these differences aren't accounted for in the analysis. Rather than just pointing out that recruiting perfect controls would be impossible, it would be more appropriate to either 1) truly incorporate these limitations into the analysis (i.e. via additional covariates), or at least 2) acknowledge the degree to which the current analysis cannot distinguish causality or differences among these confounding effects.*

We acknowledge that this is a limitation of the study, and have discussed this as a limitation in the discussion section of the manuscript.

** Likewise the concern in our second previous point (NAFLD-cirrhosis vs. NAFLD-HCC differences) was not that the comparison was not performed, but that the results - despite being very interesting - were not analyzed. This appears to still be the case, with the quantitative results relegated to the supplement with no corresponding main text. Likewise we were not able to find any quantitative or systematic comparison with the previous biomarker presented, just a qualitative mention of several shared microbes in the Discussion.*

We have further emphasised the microbiota signature of NAFLD-HCC compared to NAFLD-cirrhosis in the results, and have also made appropriate reference to both the main (Figure 2) and supplementary figures (Supplementary Figure 3) that illustrate this finding.

** The figure improvements are a great help both for visualization and for quantitative clarity. Likewise, the minor text, Methods, and reference updates are all helpful.*

We thank the reviewer for this comment.

** In the section detailing experiments stimulating PBMCs with bacterial extracts selected from "the most dysbiotic samples," was this intended to indicate selection from the least diverse (as dysbiotic and least diverse are not synonymous)?*

We agree and have accordingly corrected this sentence in the manuscript for clarity.

** In the same section, the mean (SD) for number of observed species is in the mid-3,000s vs. 5,000s for controls. This appears to be a typographical or transcribing error, but if not warrants further explanation, as even the most deeply sequenced human microbiome samples typically have species counts on the order of ~low hundreds. This elevated count is also observed in Fig. 2A. These counts might be describing strain variants or some other type of sub-species clade instead?*

We thank the reviewer for raising this to our attention. Figure 2a, and its associated reference in the manuscript was derived from alpha-diversity data output generated at an earlier 'pre-filtered' step by KrakenUniq pipeline. The figure and associated text in the manuscript have been corrected. Our data

agrees with the reviewer's assertion that human microbiome samples should have species counts in order of ~low hundreds.

** In Table 1, if no further diet information can be provided beyond simply "mixed," it could be omitted and replaced with text simply stating that these were all free-living individuals consuming ad-libitum diets.*

We have adjusted Table 1 accordingly.

** In Supplementary Fig. 4, the inclusion of essentially all taxonomic ranks (i.e. species, genus, family, etc.) in the LDA leads to a laundry list of significant associations, many of which are duplicative (Fig. 4B and clearest in clades with few branches e.g. Fusobacteriaceae). It may be useful to visualize only the species level given the metagenomic approach and lack of biologically actionable genus-level (or higher) associations.*

We respectfully request to maintain this figure as is and the data in Supplementary Fig 2 (previously Supplementary Fig 4). We agree that species level taxonomic annotation are most biologically actionable, but believe that including all taxonomic ranks will allow our results to be closely compared to studies with less comprehensive sequencing (16S rRNA).

Reviewer #3 (Remarks to the Author):

The authors adequately addressed all my concerns.

Reviewer #4 (Remarks to the Author):

I have read with great interest the revised manuscript of the study performed by Behary et al. The revised version have been improved and the authors have satisfactory addressed the majority of the comment from the reviewer #2. There is still an issue regarding the classification of the study population regarding the control group as pointed by the reviewer #2 point 1. Indeed, based upon the supplemental methods, the participants of the control group were carefully screened for other causes of liver disease than NAFLD. "In addition, among the control group 13,3% had type 2 diabetes mellitus, 16,6% had hypertension and 10% had dyslipidemia in Table 1. Hence, these participants are not "healthy" but non-NAFLD controls. This reviewer agree with the reviewer 2 and suggest to rename the "healthy group" as "non-NAFLD group" along the text, figure and Table for better accuracy as suggested initially by the reviewer 2.

We have renamed 'healthy control' to 'non-NAFLD control'. This change has been applied throughout the manuscript, all Figures, Figure Legends, Tables and Supplementary Information.